# Regret Lower Bound and Optimal Algorithm in Finite Stochastic Partial Monitoring

**Junpei Komiyama**
The University of Tokyo
junpei@komiyama.info

**Junya Honda**
The University of Tokyo
honda@stat.t.u-tokyo.ac.jp

**Hiroshi Nakagawa**
The University of Tokyo
nakagawa@dl.itc.u-tokyo.ac.jp

## Abstract

Partial monitoring is a general model for sequential learning with limited feedback formalized as a game between two players. In this game, the learner chooses an action and at the same time the opponent chooses an outcome, then the learner suffers a loss and receives a feedback signal. The goal of the learner is to minimize the total loss. In this paper, we study partial monitoring with finite actions and stochastic outcomes. We derive a logarithmic distribution-dependent regret lower bound that defines the hardness of the problem. Inspired by the DMED algorithm (Honda and Takemura, 2010) for the multi-armed bandit problem, we propose PM-DMED, an algorithm that minimizes the distribution-dependent regret. PM-DMED significantly outperforms state-of-the-art algorithms in numerical experiments. To show the optimality of PM-DMED with respect to the regret bound, we slightly modify the algorithm by introducing a hinge function (PM-DMED-Hinge). Then, we derive an asymptotically optimal regret upper bound of PM-DMED-Hinge that matches the lower bound.

## 1 Introduction

Partial monitoring is a general framework for sequential decision making problems with imperfect feedback. Many classes of problems, including prediction with expert advice [1], the multi-armed bandit problem [2], dynamic pricing [3], the dark pool problem [4], label efficient prediction [5], and linear and convex optimization with full or bandit feedback [6, 7] can be modeled as an instance of partial monitoring.

Partial monitoring is formalized as a repeated game played by two players called a learner and an opponent. At each round, the learner chooses an action, and at the same time the opponent chooses an outcome. Then, the learner observes a feedback signal from a given set of symbols and suffers some loss, both of which are deterministic functions of the selected action and outcome.

The goal of the learner is to find the optimal action that minimizes his/her cumulative loss. Alternatively, we can define the regret as the difference between the cumulative losses of the learner and the single optimal action, and minimization of the loss is equivalent to minimization of the regret. A learner with a small regret balances exploration (acquisition of information about the strategy of the opponent) and exploitation (utilization of information). The rate of regret indicates how fast the learner adapts to the problem: a linear regret indicates the inability of the learner to find the optimal action, whereas a sublinear regret indicates that the learner can approach the optimal action given sufficiently large time steps.

The study of partial monitoring is classified into two settings with respect to the assumption on the outcomes. On one hand, in the stochastic setting, the opponent chooses an outcome distribution before the game starts, and an outcome at each round is an i.i.d. sample from the distribution. On the other hand, in the adversarial setting, the opponent chooses the outcomes to maximize the regret of the learner. In this paper, we study the former setting.

## 1.1 Related work

The paper by Piccolboni and Schindelhauer [8] is one of the first to study the regret of the finite partial monitoring problem. They proposed the FeedExp3 algorithm, which attains $O(T^{3/4})$ minimax regret on some problems. This bound was later improved by Cesa-Bianchi et al. [9] to $O(T^{2/3})$, who also showed an instance in which the bound is optimal. Since then, most literature on partial monitoring has dealt with the minimax regret, which is the worst-case regret over all possible opponent's strategies. Bartók et al. [10] classified the partial monitoring problems into four categories in terms of the minimax regret: a trivial problem with zero regret, an easy problem with $\tilde{\Theta}(\sqrt{T})$ regret[1], a hard problem with $\Theta(T^{2/3})$ regret, and a hopeless problem with $\Theta(T)$ regret. This shows that the class of the partial monitoring problems is not limited to the bandit sort but also includes larger classes of problems, such as dynamic pricing. Since then, several algorithms with a $\tilde{O}(\sqrt{T})$ regret bound for easy problems have been proposed [11, 12, 13]. Among them, the Bayes-update Partial Monitoring (BPM) algorithm [13] is state-of-the-art in the sense of empirical performance.

**Distribution-dependent and minimax regret:** we focus on the distribution-dependent regret that depends on the strategy of the opponent. While the minimax regret in partial monitoring has been extensively studied, little has been known on distribution-dependent regret in partial monitoring. To the authors' knowledge, the only paper focusing on the distribution-dependent regret in finite discrete partial monitoring is the one by Bartók et al. [11], which derived $O(\log T)$ distribution-dependent regret for easy problems. In contrast to this situation, much more interest in the distribution-dependent regret has been shown in the field of multi-armed bandit problems. Upper confidence bound (UCB), the most well-known algorithm for the multi-armed bandits, has a distribution-dependent regret bound [2, 14], and algorithms that minimize the distribution-dependent regret (e.g., KL-UCB) has been shown to perform better than ones that minimize the minimax regret (e.g., MOSS), even in instances in which the distributions are hard to distinguish (e.g., Scenario 2 in Garivier et al. [15]). Therefore, in the field of partial monitoring, we can expect that an algorithm that minimizes the distribution-dependent regret would perform better than the existing ones.

**Contribution:** the contributions of this paper lie in the following three aspects. First, we derive the regret lower bound: in some special classes of partial monitoring (e.g., multi-armed bandits), an $O(\log T)$ regret lower bound is known to be achievable. In this paper, we further extend this lower bound to obtain a regret lower bound for general partial monitoring problems. Second, we propose an algorithm called Partial Monitoring DMED (PM-DMED). We also introduce a slightly modified version of this algorithm (PM-DMED-Hinge) and derive its regret bound. PM-DMED-Hinge is the first algorithm with a logarithmic regret bound for hard problems. Moreover, for both easy and hard problems, it is the first algorithm with the optimal constant factor on the leading logarithmic term. Third, performances of PM-DMED and existing algorithms are compared in numerical experiments. Here, the partial monitoring problems consisted of three specific instances of varying difficulty. In all instances, PM-DMED significantly outperformed the existing methods when a number of rounds is large. The regret of PM-DMED on these problems quickly approached the theoretical lower bound.

## 2 Problem Setup

This paper studies the finite stochastic partial monitoring problem with $N$ actions, $M$ outcomes, and $A$ symbols. An instance of the partial monitoring game is defined by a loss matrix $L = (l_{i,j}) \in \mathbb{R}^{N \times M}$ and a feedback matrix $H = (h_{i,j}) \in [A]^{N \times M}$, where $[A] = \{1, 2, \ldots, A\}$. At the beginning, the learner is informed of $L$ and $H$. At each round $t = 1, 2, \ldots, T$, a learner selects an action $i(t) \in [N]$, and at the same time an opponent selects an outcome $j(t) \in [M]$. The learner

suffers loss $l_{i(t),j(t)}$, which he/she cannot observe: the only information the learner receives is the signal $h_{i(t),j(t)} \in [A]$. We consider a stochastic opponent whose strategy for selecting outcomes is governed by the opponent's strategy $p^* \in \mathcal{P}_M$, where $\mathcal{P}_M$ is a set of probability distributions over an $M$-ary outcome. The outcome $j(t)$ of each round is an i.i.d. sample from $p^*$.

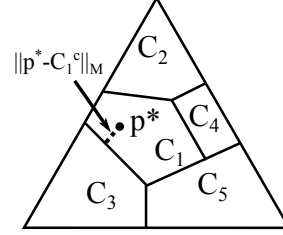

The goal of the learner is to minimize the cumulative loss over $T$ rounds. Let the optimal action be the one that minimizes the loss in expectation, that is, $i^* = \arg\min_{i \in [N]} L_i^\top p^*$, where $L_i$ is the $i$-th row of $L$. Assume that $i^*$ is unique. Without loss of generality, we can assume that $i^* = 1$. Let $\Delta_i = (L_i - L_1)^\top p^* \in [0, \infty)$ and $N_i(t)$ be the number of rounds before the $t$-th in which action $i$ is selected. The performance of the algorithm is measured by the (pseudo) regret,

$$\text{Regret}(T) = \sum_{t=1}^{T} \Delta_{i(t)} = \sum_{i \in [N]} \Delta_i N_i(T+1),$$

Figure 1: Cell decomposition of a partial monitoring instance with $M = 3$.

which is the difference between the expected loss of the learner and the optimal action. It is easy to see that minimizing the loss is equivalent to minimizing the regret. The expectation of the regret measures the performance of an algorithm that the learner uses.

For each action $i \in [N]$, let $\mathcal{C}_i$ be the set of opponent strategies for which action $i$ is optimal:

$$\mathcal{C}_i = \{q \in \mathcal{P}_M : \forall_{j \neq i} (L_i - L_j)^\top q \leq 0\}.$$

We call $\mathcal{C}_i$ the optimality cell of action $i$. Each optimality cell is a convex closed polytope. Furthermore, we call the set of optimality cells $\{\mathcal{C}_1, \ldots, \mathcal{C}_N\}$ the cell decomposition as shown in Figure 1. Let $\mathcal{C}_i^c = \mathcal{P}_M \setminus \mathcal{C}_i$ be the set of strategies with which action $i$ is not optimal.

The signal matrix $S_i \in \{0,1\}^{A \times M}$ of action $i$ is defined as $(S_i)_{k,j} = \mathbb{1}[h_{i,j} = k]$, where $\mathbb{1}[X] = 1$ if $X$ is true and 0 otherwise. The signal matrix defined here is slightly different from the one in the previous papers (e.g., Bartók et al. [10]) in which the number of rows of $S_i$ is the number of the different symbols in the $i$-th row of $H$. The advantage in using the definition here is that, $S_i p^* \in \mathbb{R}^A$ is a probability distribution over symbols that the algorithm observes when it selects an action $i$. Examples of signal matrices are shown in Section 5. An instance of partial monitoring is *globally observable* if for all pairs $i, j$ of actions, $L_i - L_j \in \oplus_{k \in [N]} \text{Im} S_k^\top$. In this paper, we exclusively deal with globally observable instances: in view of the minimax regret, this includes trivial, easy, and hard problems.

## 3 Regret Lower Bound

A good algorithm should work well against any opponent's strategy. We extend this idea by introducing the notion of strong consistency: a partial monitoring algorithm is strongly consistent if it satisfies $\mathbb{E}[\text{Regret}(T)] = o(T^a)$ for any $a > 0$ and $p \in \mathcal{P}_M$ given $L$ and $H$.

In the context of the multi-armed bandit problem, Lai and Robbins [2] derived the regret lower bound of a strongly consistent algorithm: an algorithm must select each arm $i$ until its number of draws $N_i(t)$ satisfies $\log t \lesssim N_i(t) d(\theta_i \| \theta_1)$, where $d(\theta_i \| \theta_1)$ is the KL divergence between the two one-parameter distributions from which the rewards of action $i$ and the optimal action are generated. Analogously, in the partial monitoring problem, we can define the minimum number of observations.

**Lemma 1.** *For sufficiently large $T$, a strongly consistent algorithm satisfies:*

$$\forall_{q \in \mathcal{C}_1^c} \sum_{i \in [N]} \mathbb{E}[N_i(T)] D(p_i^* \| S_i q) \geq \log T - o(\log T),$$

*where $p_i^* = S_i p^*$ and $D(p \| q) = \sum_i (p)_i \log ((p)_i / (q)_i)$ is the KL divergence between two discrete distributions, in which we define $0 \log 0 / 0 = 0$.*

Lemma 1 can be interpreted as follows: for each round $t$, consistency requires the algorithm to make sure that the possible risk that action $i \neq 1$ is optimal is smaller than $1/t$. Large deviation principle [16] states that, the probability that an opponent with strategy $q$ behaves like $p^*$ is

roughly $\exp\left(-\sum_i N_i(t)D(p_i^*\|S_i q)\right)$. Therefore, we need to continue exploration of the actions until $\sum_i N_i(t)D(p_i^*\|S_i q) \sim \log t$ holds for any $q \in \mathcal{C}_1^c$ to reduce the risk to $\exp\left(-\log t\right) = 1/t$.

The proof of Lemma 1 is in Appendix B in the supplementary material. Based on the technique used in Lai and Robbins [2], the proof considers a modified game in which another action $i \neq 1$ is optimal. The difficulty in proving the lower bound in partial monitoring lies in that, the feedback structure can be quite complex: for example, to confirm the superiority of action 1 over 2, one might need to use the feedback from action $3 \notin \{1, 2\}$. Still, we can derive the lower bound by utilizing the consistency of the algorithm in the original and modified games.

We next derive a lower bound on the regret based on Lemma 1. Note that, the expectation of the regret can be expressed as $\mathbb{E}[\mathrm{Regret}(T)] = \sum_{i \neq 1} \mathbb{E}[N_i(t)](L_i - L_1)^\top p^*$. Let

$$\mathcal{R}_j(\{p_i\}) = \left\{ \{r_i\}_{i \neq j} \in [0, \infty)^{N-1} : \inf_{q \in \mathrm{cl}(\mathcal{C}_j^c):p_j = S_j q} \sum_i r_i D(p_i\|S_i q) \geq 1 \right\},$$

where $\mathrm{cl}(\cdot)$ denotes a closure. Moreover, let

$$C_j^*(p, \{p_i\}) = \inf_{r_i \in \mathcal{R}_j(\{p_i\})} \sum_{i \neq j} r_i(L_i - L_j)^\top p,$$

the optimal solution of which is

$$\mathcal{R}_j^*(p, \{p_i\}) = \left\{ \{r_i\}_{i \neq j} \in \mathcal{R}_j(\{p_i\}) : \sum_{i \neq j} r_i(L_i - L_j)^\top p = C_j^*(p, \{p_i\}) \right\}.$$

The value $C_1^*(p^*, \{p_i^*\}) \log T$ is the possible minimum regret for observations such that the minimum divergence of $p^*$ from any $q \in \mathcal{C}_1^c$ is larger than $\log T$. Using Lemma 1 yields the following regret lower bound:

**Theorem 2.** *The regret of a strongly consistent algorithm is lower bounded as:*

$$\mathbb{E}[\mathrm{Regret}(T)] \geq C_1^*(p^*, \{p_i^*\}) \log T - \mathrm{o}(\log T).$$

From this theorem, we can naturally measure the harshness of the instance by $C_1^*(p^*, \{p_i^*\})$, whereas the past studies (e.g., Vanchinathan et al. [13]) ambiguously define the harshness as the closeness to the boundary of the cells. Furthermore, we show in Lemma 5 in the Appendix that $C_1^*(p^*, \{p_i^*\}) = \mathrm{O}(N/\|p^* - \mathcal{C}_1^c\|_M^2)$: the regret bound has at most quadratic dependence on $\|p^* - \mathcal{C}_1^c\|_M$, which is defined in Appendix D as the closeness of $p^*$ to the boundary of the optimal cell.

## 4  PM-DMED Algorithm

In this section, we describe the partial monitoring deterministic minimum empirical divergence (PM-DMED) algorithm, which is inspired by DMED [17] for solving the multi-armed bandit problem. Let $\hat{p}_i(t) \in [0, 1]^A$ be the empirical distribution of the symbols under the selection of action $i$. Namely, the $k$-th element of $\hat{p}_i(t)$ is $(\sum_{t'=1}^{t-1} \mathbb{1}[i(t') = i \cap h_{i(t'),j(t')} = k])/(\sum_{t'=1}^{t-1} \mathbb{1}[i(t') = i])$. We sometimes omit $t$ from $\hat{p}_i$ when it is clear from the context. Let the empirical divergence of $q \in \mathcal{P}_M$ be $\sum_{i \in [N]} N_i(t)D(\hat{p}_i(t)\|S_i q)$, the exponential of which can be considered as a likelihood that $q$ is the opponent's strategy.

The main routine of PM-DMED is in Algorithm 1. At each loop, the actions in the current list $Z_C$ are selected once. The list for the actions in the next loop $Z_N$ is determined by the subroutine in Algorithm 2. The subroutine checks whether the empirical divergence of each point $q \in \mathcal{C}_1^c$ is larger than $\log t$ or not (Eq. (3)). If it is large enough, it exploits the current information by selecting $\hat{i}(t)$, the optimal action based on the estimation $\hat{p}(t)$ that minimizes the empirical divergence. Otherwise, it selects the actions with the number of observations below the minimum requirement for making the empirical divergence of each suboptimal point $q \in \mathcal{C}_1^c$ larger than $\log t$.

Unlike the $N$-armed bandit problem in which a reward is associated with an action, in the partial monitoring problem, actions, outcomes, and feedback signals can be intricately related. Therefore, we need to solve a non-trivial optimization to run PM-DMED. Later in Section 5, we discuss a practical implementation of the optimization.

**Algorithm 1** Main routine of PM-DMED and PM-DMED-Hinge

1: **Initialization:** select each action once.
2: $Z_C, Z_R \leftarrow [N], Z_N \leftarrow \emptyset$.
3: **while** $t \leq T$ **do**
4:    **for** $i(t) \in Z_C$ in an arbitrarily fixed order **do**
5:       Select $i(t)$, and receive feedback.
6:       $Z_R \leftarrow Z_R \setminus \{i(t)\}$.
7:       Add actions to $Z_N$ in accordance with
$$\begin{cases} \text{Algorithm 2} & \text{(PM-DMED)} \\ \text{Algorithm 3} & \text{(PM-DMED-Hinge)} \end{cases}.$$
8:       $t \leftarrow t + 1$.
9:    **end for**
10:    $Z_C, Z_R \leftarrow Z_N, Z_N \leftarrow \emptyset$.
11: **end while**

**Algorithm 2** PM-DMED subroutine for adding actions to $Z_N$ (without duplication).

1: **Parameter:** $c > 0$.
2: Compute an arbitrary $\hat{p}(t)$ such that
$$\hat{p}(t) \in \arg\min_q \sum_i N_i(t) D(\hat{p}_i(t) \| S_i q) \quad (1)$$
and let $\hat{i}(t) = \arg\min_i L_i^\top \hat{p}(t)$.
3: If $\hat{i}(t) \notin Z_R$ then put $\hat{i}(t)$ into $Z_N$.
4: If there are actions $i \notin Z_R$ such that
$$N_i(t) < c\sqrt{\log t} \quad (2)$$
then put them into $Z_N$.
5: If
$$\{N_i(t)/\log t\}_{i \neq \hat{i}(t)} \notin \mathcal{R}_{\hat{i}(t)}(\{\hat{p}_i(t)\}) \quad (3)$$
then compute some
$$\{r_i^*\}_{i \neq \hat{i}(t)} \in \mathcal{R}_{\hat{i}(t)}^*(\hat{p}(t), \{\hat{p}_i(t)\}) \quad (4)$$
and put all actions $i$ such that $i \notin Z_R$ and $r_i^* > N_i(t)/\log t$ into $Z_N$.

**Necessity of $\sqrt{\log T}$ exploration:** PM-DMED tries to observe each action to some extent (Eq. (2)), which is necessary for the following reason: consider a four-state game characterized by

$$L = \begin{pmatrix} 0 & 1 & 1 & 0 \\ 10 & 1 & 0 & 0 \\ 10 & 0 & 1 & 0 \\ 11 & 11 & 11 & 11 \end{pmatrix}, \quad H = \begin{pmatrix} 1 & 1 & 1 & 1 \\ 1 & 2 & 2 & 3 \\ 1 & 2 & 2 & 3 \\ 1 & 1 & 2 & 2 \end{pmatrix}, \text{ and } \quad p^* = (0.1, 0.2, 0.3, 0.4)^\top.$$

The optimal action here is action 1, which does not yield any useful information. By using action 2, one receives three kinds of symbols from which one can estimate $(p^*)_1$, $(p^*)_2 + (p^*)_3$, and $(p^*)_4$, where $(p^*)_j$ is the $j$-th component of $p^*$. From this, an algorithm can find that $(p^*)_1$ is not very small and thus the expected loss of actions 2 and 3 is larger than that of action 1. Since the feedback of actions 2 and 3 are the same, one may also use action 3 in the same manner. However, the loss per observation is 1.2 and 1.3 for actions 2 and 3, respectively, and thus it is better to use action 2. This difference comes from the fact that $(p^*)_2 = 0.2 < 0.3 = (p^*)_3$. Since an algorithm does not know $p^*$ beforehand, it needs to observe action 4, the only source for distinguishing $(p^*)_2$ from $(p^*)_3$. Yet, an optimal algorithm cannot select it more than $\Omega(\log T)$ times because it affects the $O(\log T)$ factor in the regret. In fact, $O((\log T)^a)$ observations of action 4 with some $a > 0$ are sufficient to be convinced that $(p^*)_2 < (p^*)_3$ with probability $1 - o(1/T^{\text{poly}(a)})$. For this reason, PM-DMED selects each action $\sqrt{\log t}$ times.

## 5 Experiment

Following Bartók et al. [11], we compared the performances of algorithms in three different games: the four-state game (Section 4), a three-state game and dynamic pricing. Experiments on the $N$-armed bandit game was also done, and the result is shown in Appendix C.1 .

The three-state game, which is classified as easy in terms of the minimax regret, is characterized by:

$$L = \begin{pmatrix} 1 & 1 & 0 \\ 0 & 1 & 1 \\ 1 & 0 & 1 \end{pmatrix} \quad \text{and} \quad H = \begin{pmatrix} 1 & 2 & 2 \\ 2 & 1 & 2 \\ 2 & 2 & 1 \end{pmatrix}.$$

The signal matrices of this game are,

$$S_1 = \begin{pmatrix} 1 & 0 & 0 \\ 0 & 1 & 1 \end{pmatrix}, \quad S_2 = \begin{pmatrix} 0 & 1 & 0 \\ 1 & 0 & 1 \end{pmatrix}, \text{ and } S_3 = \begin{pmatrix} 0 & 0 & 1 \\ 1 & 1 & 0 \end{pmatrix}.$$

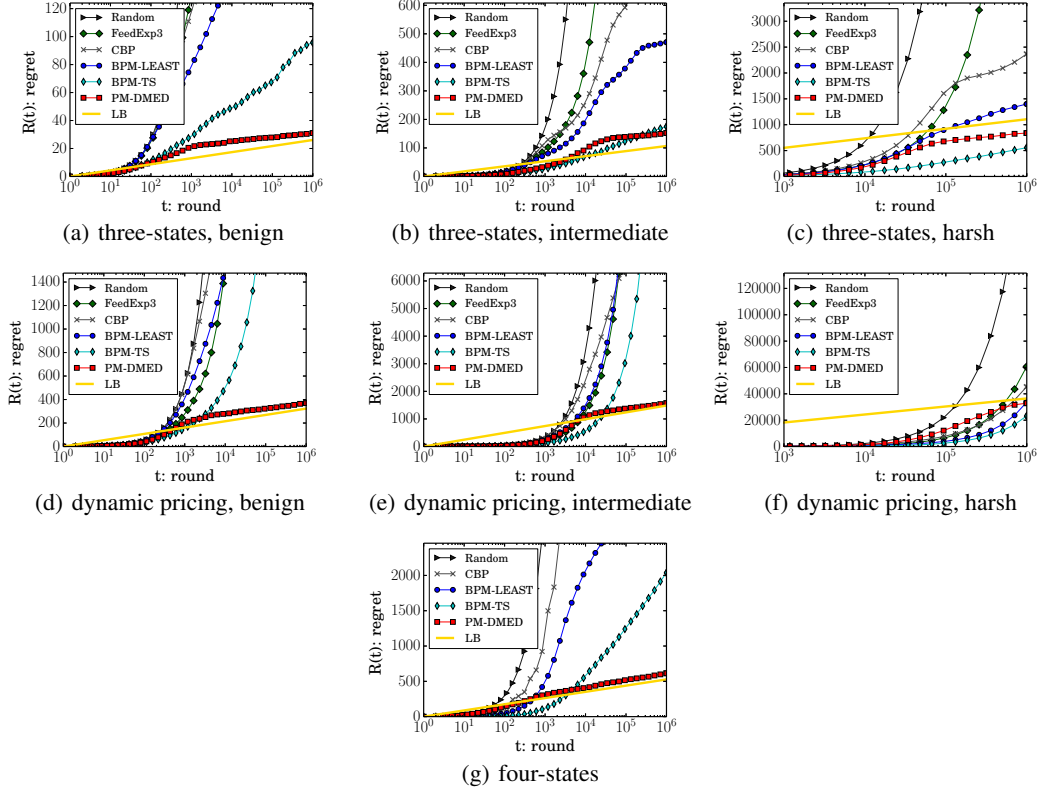

Figure 2: Regret-round semilog plots of algorithms. The regrets are averaged over 100 runs. LB is the asymptotic regret lower bound of Theorem 2.

Dynamic pricing, which is classified as hard in terms of the minimax regret, is a game that models a repeated auction between a seller (learner) and a buyer (opponent). At each round, the seller sets a price for a product, and at the same time, the buyer secretly sets a maximum price he is willing to pay. The signal is "buy" or "no-buy", and the seller's loss is either a given constant (no-buy) or the difference between the buyer's and the seller's prices (buy). The loss and feedback matrices are:

$$
L = \begin{pmatrix} 0 & 1 & \dots & N-1 \\ c & 0 & \dots & N-2 \\ \vdots & \ddots & \ddots & \vdots \\ c & \dots & c & 0 \end{pmatrix} \quad \text{and} \quad H = \begin{pmatrix} 2 & 2 & \dots & 2 \\ 1 & 2 & \dots & 2 \\ \vdots & \ddots & \ddots & \vdots \\ 1 & \dots & 1 & 2 \end{pmatrix},
$$

where signals 1 and 2 correspond to no-buy and buy. The signal matrix of action $i$ is

$$
S_i = \begin{pmatrix} \overbrace{1 \quad \dots \quad 1}^{i-1} & \overbrace{0 \quad \dots \quad 0}^{M-i+1} \\ 0 \quad \dots \quad 0 & 1 \quad \dots \quad 1 \end{pmatrix}.
$$

Following Bartók et al. [11], we set $N = 5, M = 5$, and $c = 2$.

In our experiments with the three-state game and dynamic pricing, we tested three settings regarding the harshness of the opponent: at the beginning of a simulation, we sampled 1,000 points uniformly at random from $\mathcal{P}_M$, then sorted them by $C_1^*(p^*, \{p_i^*\})$. We chose the top 10%, 50%, and 90% harshest ones as the opponent's strategy in the harsh, intermediate, and benign settings, respectively.

We compared Random, FeedExp3 [8], CBP [11] with $\alpha = 1.01$, BPM-LEAST, BPM-TS [13], and PM-DMED with $c = 1$. Random is a naive algorithm that selects an action uniformly random. FeedExp3 requires a matrix $G$ such that $H^\top G = L^\top$, and thus one cannot apply it to the four-state game. CBP is an algorithm of logarithmic regret for easy games. The parameters $\eta$ and $f(t)$ of CBP were set in accordance with Theorem 1 in their paper. BPM-LEAST is a Bayesian algorithm with $\tilde{O}(\sqrt{T})$ regret for easy games, and BPM-TS is a heuristic of state-of-the-art performance. The priors of two BPMs were set to be uninformative to avoid a misspecification, as recommended in their paper.

---

**Algorithm 3** PM-DMED-Hinge subroutine for adding actions to $Z_N$ (without duplication).

1: **Parameters:** $c > 0$, $f(n) = bn^{-1/2}$ for $b > 0$, $\alpha(t) = a/(\log \log t)$ for $a > 0$.
2: Compute arbitrary $\hat{p}(t)$ which satisfies

$$\hat{p}(t) \in \arg\min_q \sum_i N_i(t)(D(\hat{p}_i(t)\|S_i q) - f(N_i(t)))_+ \tag{5}$$

  and let $\hat{i}(t) = \arg\min_i L_i^\top \hat{p}(t)$.
3: If $\hat{i}(t) \notin Z_R$ then put $\hat{i}(t)$ into $Z_N$.
4: If

$$\hat{p}(t) \notin \mathcal{C}_{\hat{i}(t),\alpha(t)} \tag{6}$$

  or there exists an action $i$ such that

$$D(\hat{p}_i(t)\|S_i\hat{p}(t)) > f(N_i(t)) \tag{7}$$

  then put all actions $i \notin Z_R$ into $Z_N$.
5: If there are actions $i$ such that
$$N_i(t) < c\sqrt{\log t} \tag{8}$$
  then put the actions not in $Z_R$ into $Z_N$.
6: If
$$\{N_i(t)/\log t\}_{i \neq \hat{i}(t)} \notin \mathcal{R}_{\hat{i}(t)}(\{\hat{p}_i(t), f(N_i(t))\}) \tag{9}$$

  then compute some

$$\{r_i^*\}_{i \neq \hat{i}(t)} \in \mathcal{R}_{\hat{i}(t)}^*(\hat{p}(t), \{\hat{p}_i(t), f(N_i(t))\}) \tag{10}$$

  and put all actions such that $i \notin Z_R$ and $r_i^* > N_i(t)/\log t$ into $Z_N$. If such $r_i^*$ is infeasible then put all action $i \notin Z_R$ into $Z_N$.

---

The computation of $\hat{p}(t)$ in (1) and the evaluation of the condition in (3) involve convex optimizations, which were done with Ipopt [18]. Moreover, obtaining $\{r_i^*\}$ in (4) is classified as a linear semi-infinite programming (LSIP) problem, a linear programming (LP) with finitely many variables and infinitely many constraints. Following the optimization of BPM-LEAST [13], we resorted to a finite sample approximation and used the Gurobi LP solver [19] in computing $\{r_i^*\}$: at each round, we sampled 1,000 points from $\mathcal{P}_M$, and relaxed the constraints on the samples. To speed up the computation, we skipped these optimizations in most rounds with large $t$ and used the result of the last computation. The computation of the coefficient $C_1^*(p^*, \{p_i^*\})$ of the regret lower bound (Theorem 2) is also an LSIP, which was approximated by 100,000 sample points from $\mathcal{C}_1^c$.

The experimental results are shown in Figure 2. In the four-state game and the other two games with an easy or intermediate opponent, PM-DMED outperforms the other algorithms when the number of rounds is large. In particular, in the dynamic pricing game with an intermediate opponent, the regret of PM-DMED at $T = 10^6$ is ten times smaller than those of the other algorithms. Even in the harsh setting in which the minimax regret matters, PM-DMED has some advantage over all algorithms except for BPM-TS. With sufficiently large $T$, the slope of an optimal algorithm should converge to LB. In all games and settings, the slope of PM-DMED converges to LB, which is empirical evidence of the optimality of PM-DMED.

## 6 Theoretical Analysis

Section 5 shows that the empirical performance of PM-DMED is very close to the regret lower bound in Theorem 2. Although the authors conjecture that PM-DMED is optimal, it is hard to analyze PM-DMED. The technically hardest part arises from the case in which the divergence of each action is small but not yet fully converged. To circumvent this difficulty, we can introduce a discount factor. Let

$$\mathcal{R}_j(\{p_i, \delta_i\}) = \left\{ \{r_i\}_{i \neq j} \in [0, \infty)^{N-1} : \inf_{q \in \mathrm{cl}(\mathcal{C}_j^c): D(p_j\|S_j q) \leq \delta_j} \sum_i r_i (D(p_i\|S_i q) - \delta_i)_+ \geq 1 \right\}, \tag{11}$$

where $(X)_+ = \max(X, 0)$. Note that $\mathcal{R}_j(\{p_i, \delta_i\})$ in (11) is a natural generalization of $\mathcal{R}_j(\{p_i\})$ in Section 4 in the sense that $\mathcal{R}_j(\{p_i, 0\}) = \mathcal{R}_j(\{p_i\})$. Event $\{N_i(t)/\log t\}_{i \neq 1} \in \mathcal{R}_1(\{\hat{p}_i(t), \delta_i\})$ means that the number of observations $\{N_i(t)\}$ is enough to ensure that the "$\{\delta_i\}$-discounted" empirical divergence of each $q \in \mathcal{C}_1^c$ is larger than $\log t$. Analogous to $\mathcal{R}_j(\{p_i, \delta_i\})$, we define

$$C_j^*(p, \{p_i, \delta_i\}) = \inf_{\{r_i\}_{i \neq j} \in \mathcal{R}_j(\{p_i, \delta_i\}))} \sum_{i \neq j} r_i (L_j - L_i)^\top p$$

and its optimal solution by

$$\mathcal{R}_j^*(p, \{p_i, \delta_i\}) = \left\{ \{r_i\}_{i \neq j} \in \mathcal{R}_j(\{p_i, \delta_i\}) : \sum_{i \neq j} r_i (L_j - L_i)^\top p = C_j^*(p, \{p_i, \delta_i\}) \right\}.$$

We also define $\mathcal{C}_{i,\alpha} = \{p \in \mathcal{P}_M : L_i^\top p + \alpha \leq \min_{j \neq i} L_j^\top p\}$, the optimal region of action $i$ with margin. PM-DMED-Hinge shares the main routine of Algorithm 1 with PM-DMED and lists the next actions by Algorithm 3. Unlike PM-DMED, it (i) discounts $f(N_i(t))$ from the empirical divergence $D(\hat{p}_i(t)\|S_i q)$. Moreover, (ii) when $\hat{p}(t)$ is close to the cell boundary, it encourages more exploration to identify the cell it belongs to by Eq. (6).

**Theorem 3.** *Assume that the following regularity conditions hold for $p^*$. (1) $\mathcal{R}_1^*(p, \{p_i, \delta_i\})$ is unique at $p = p^*, p_i = S_i p^*, \delta_i = 0$. Moreover, (2) for $\mathcal{S}_\delta = \{q : D(p_1^*\|S_1 q) \leq \delta\}$, it holds that $\mathrm{cl}(\mathrm{int}(\mathcal{C}_1^c) \cap \mathcal{S}_\delta) = \mathrm{cl}(\mathrm{cl}(\mathcal{C}_1^c) \cap \mathcal{S}_\delta)$ for all $\delta \geq 0$ in some neighborhood of $\delta = 0$, where $\mathrm{cl}(\cdot)$ and $\mathrm{int}(\cdot)$ denote the closure and the interior, respectively. Then,*

$$\mathbb{E}[\mathrm{Regret}(T)] \leq C_1^*(p^*, \{p_i^*\}) \log T + \mathrm{o}(\log T).$$

We prove this theorem in Appendix D . Recall that $\mathcal{R}_1^*(p, \{\hat{p}_i(t), \delta_i\})$ is the set of optimal solutions of an LSIP. In this problem, KKT conditions and the duality theorem apply as in the case of finite constraints; thus, we can check whether Condition 1 holds or not for each $p^*$ (see, e.g., Ito et al. [20] and references therein). Condition 2 holds in most cases, and an example of an exceptional case is shown in Appendix A.

Theorem 3 states that PM-DMED-Hinge has a regret upper bound that matches the lower bound of Theorem 2.

**Corollary 4.** *(Optimality in the $N$-armed bandit problem) In the $N$-armed Bernoulli bandit problem, the regularity conditions in Theorem 3 always hold. Moreover, the coefficient of the leading logarithmic term in the regret bound of the partial monitoring problem is equal to the bound given in Lai and Robbins [2]. Namely, $C_1^*(p^*, \{p_i^*\}) = \sum_{i \neq 1}^{N}(\Delta_i/d(\mu_i\|\mu_1))$, where $d(p\|q) = p \log(p/q) + (1-p) \log((1-p)/(1-q))$ is the KL-divergence between Bernoulli distributions.*

Corollary 4, which is proven in Appendix C, states that PM-DMED-Hinge attains the optimal regret of the $N$-armed bandit if we run it on an $N$-armed bandit game represented as partial monitoring.

**Asymptotic analysis:** it is Theorem 6 where we lose the finite-time property. This theorem shows the continuity of the optimal solution set $\mathcal{R}_1^*(p, \{p_i, \delta_i\})$ of $C_j^*(p, \{p_j\})$, which does not mention how close $\mathcal{R}_1^*(p, \{p_i, \delta_i\})$ is to $\mathcal{R}_1^*(p^*, \{p_i^*, 0\})$ if $\max\{\|p - p^*\|_M, \max_i \|p_i - p_i^*\|_M, \max_i \delta_i\} \leq \delta$ for given $\delta$. To obtain an explicit bound, we need *sensitivity analysis*, the theory of the robustness of the optimal value and the solution for small deviations of its parameters (see e.g., Fiacco [21]). In particular, the optimal solution of partial monitoring involves an infinite number of constraints, which makes the analysis quite hard. For this reason, we will not perform a finite-time analysis. Note that, the $N$-armed bandit problem is a special instance in which we can avoid solving the above optimization and a finite-time optimal bound is known.

**Necessity of the discount factor:** we are not sure whether discount factor $f(n)$ in PM-DMED-Hinge is necessary or not. We also empirically tested PM-DMED-Hinge: although it is better than the other algorithms in many settings, such as dynamic pricing with an intermediate opponent, it is far worse than PM-DMED. We found that our implementation, which uses the Ipopt nonlinear optimization solver, was sometimes inaccurate at optimizing (5): there were some cases in which the true $p^*$ satisfies $\forall_{i \in [N]} D(\hat{p}_i(t)\|S_i p^*) - f(N_i(t)) = 0$, while the solution $\hat{p}(t)$ we obtained had non-zero hinge values. In this case, the algorithm lists all actions from (7), which degrades performance. Determining whether the discount factor is essential or not is our future work.

## Acknowledgements

The authors gratefully acknowledge the advice of Kentaro Minami and sincerely thank the anonymous reviewers for their useful comments. This work was supported in part by JSPS KAKENHI Grant Number 15J09850 and 26106506.

## Footnotes

[1]Note that $\tilde{\Theta}$ ignores a polylog factor.

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
