[Supplementary Material]

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

(a) three-states, benign     (b) three-states, intermediate     (c) three-states, harsh

(d) dynamic pricing, benign     (e) dynamic pricing, intermediate     (f) dynamic pricing, harsh

(g) four-states

Figure 2: Regret-round semilog plots of algorithms. The regrets are averaged over 100 runs. LB is the asymptotic regret lower bound of Theorem 2.

Dynamic pricing, which is classified as hard in terms of the minimax regret, is a game that models a repeated auction between a seller (learner) and a buyer (opponent). At each round, the seller sets a price for a product, and at the same time, the buyer secretly sets a maximum price he is willing to pay. The signal is "buy" or "no-buy", and the seller's loss is either a given constant (no-buy) or the difference between the buyer's and the seller's prices (buy). The loss and feedback matrices are:

$$
L = \begin{pmatrix} 0 & 1 & \dots & N-1 \\ c & 0 & \dots & N-2 \\ \vdots & \ddots & \ddots & \vdots \\ c & \dots & c & 0 \end{pmatrix} \quad \text{and} \quad H = \begin{pmatrix} 2 & 2 & \dots & 2 \\ 1 & 2 & \dots & 2 \\ \vdots & \ddots & \ddots & \vdots \\ 1 & \dots & 1 & 2 \end{pmatrix},
$$

where signals 1 and 2 correspond to no-buy and buy. The signal matrix of action $i$ is

$$
S_i = \left( \overbrace{\begin{matrix} 1 & \dots & 1 \end{matrix}}^{i-1} \overbrace{\begin{matrix} 0 & \dots & 0 \\ 1 & \dots & 1 \end{matrix}}^{M-i+1} \right).
$$

$$

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

Figure 3: A corner case.

## A  Case in which Condition 2 Does Not Hold

Figure 3 is an example that Theorem 3 does not cover. The dotted line is $\{q : p_1^* = S_1 q\}$, which (accidentally) coincides with a line that makes the convex polytope of $\mathcal{C}_1^c$. In this case, Condition 2 does not hold because $\mathrm{int}(\mathcal{C}_1^c) \cap \mathcal{S}_0 = \emptyset$ whereas $\mathrm{cl}(\mathcal{C}_1^c) \cap \mathcal{S}_0 \neq \emptyset$ (two starred points), which means that a slight modification of $p^*$ changes the set of cells that intersects with the dotted line discontinuously. We exclude these unusual cases for the ease of analysis.

The authors consider that it is quite hard to give the optimal regret bound without such regularity conditions. In fact, many regularity conditions are assumed in Graves and Lai [22], where another generalization of the bandit problem is considered and the regret lower bound is expressed in terms of LSIP. In this paper, the regularity conditions are much simplified by the continuity argument in Theorem 6 but it remains an open problem to fully remove them.

## B  Proof: Regret Lower Bound

In this section, we prove Lemma 1 and Theorem 2.

*Proof of Lemma 1.* The technique here is mostly inspired from Theorem 1 in Lai and Robbins [2]. The use of a $\sqrt{T}$ term is inspired from Kaufmann et al. [23]. Let $p' \in \mathrm{int}(\mathcal{C}_1^c)$ and $i' \neq 1$ be the optimal action under the opponent's strategy $p'$. We consider a modified partial monitoring game with its opponent's strategy is $p'$.

**Notation:** Let $\hat{X}_i^m \in [A]$ is the signal of the $m$-th observation of action $i$. Let

$$\widehat{\mathrm{KL}}_i(n) = \sum_{m=1}^n \log \left( \frac{(S_i p^*)_{\hat{X}_i^m}}{(S_i p')_{\hat{X}_i^m}} \right),$$

and $\widehat{\mathrm{KL}} = \sum_{i \in [N]} \widehat{\mathrm{KL}}_i(N_i(T))$. Let $\mathbb{P}'$ and $\mathbb{E}'$ be the probability and the expectation with respect to the modified game, respectively. Then, for any event $\mathcal{E}$,

$$\mathbb{P}'[\mathcal{E}] = \mathbb{E}\left[ \mathbb{1}[\mathcal{E}] \exp\left( -\widehat{\mathrm{KL}} \right) \right] \tag{12}$$

holds. Now, let us define the following events:

$$\mathcal{D}_1 = \left\{ \sum_{i \in [N]} N_i(T) D(S_i p^* \| S_i p') < (1 - \epsilon) \log T, N_{i'}(T) < \sqrt{T} \right\},$$
$$\mathcal{D}_2 = \left\{ \widehat{\mathrm{KL}} \leq \left( 1 - \frac{\epsilon}{2} \right) \log T \right\},$$
$$\mathcal{D}_{12} = \mathcal{D}_1 \cap \mathcal{D}_2,$$
$$\mathcal{D}_{1\backslash 2} = \mathcal{D}_1 \cap \mathcal{D}_2^c.$$

**First step** ($\Pr[\mathcal{D}_{12}] = \mathrm{o}(1)$)**:** from (12),
$$\mathbb{P}'[\mathcal{D}_{12}] \geq \mathbb{E}\left[\mathbb{1}[\mathcal{D}_{12}]\exp\left(-\left(1-\frac{\epsilon}{2}\right)\log T\right)\right] = T^{-(1-\epsilon/2)}\Pr[\mathcal{D}_{12}].$$
By using this we have
$$\begin{aligned}
\Pr[\mathcal{D}_{12}] &\leq T^{(1-\epsilon/2)}\mathbb{P}'[\mathcal{D}_{12}]\\
&\leq T^{(1-\epsilon/2)}\mathbb{P}'\left[N_{i'}(T) < \sqrt{T}\right]\\
&= T^{(1-\epsilon/2)}\mathbb{P}'\left[T - N_{i'}(T) > T - \sqrt{T}\right]\\
&\leq T^{(1-\epsilon/2)}\frac{\mathbb{E}'[T - N_{i'}(T)]}{T - \sqrt{T}} \qquad \text{(by the Markov inequality).} \qquad (13)
\end{aligned}$$
Since this algorithm is strongly consistent, $\mathbb{E}'[T - N_{i'}(T)] \to \mathrm{o}(T^a)$ for any $a > 0$. Therefore, the RHS of the last line of (13) is $\mathrm{o}(T^{a-\epsilon/2})$, which, by choosing sufficiently small $a$, converges to zero as $T \to \infty$. In summary, $\Pr[\mathcal{D}_{12}] = \mathrm{o}(1)$.

**Second step** ($\Pr[\mathcal{D}_{1\backslash 2}] = \mathrm{o}(1)$)**:** we have

$\Pr[\mathcal{D}_{1\backslash 2}]$
$$= \Pr\left[\sum_{i\in[N]} N_i(T)D(S_i p^*\|S_i p') < (1-\epsilon)\log T, N_{i'}(T) < \sqrt{T}, \sum_{i\in[N]} \widehat{\mathrm{KL}}_i(N_i(T)) > \left(1-\frac{\epsilon}{2}\right)\log T\right].$$
Note that
$$\max_{1\leq n\leq N} \widehat{\mathrm{KL}}_i(n) = \max_{1\leq n\leq N}\sum_{m=1}^{n}\log\left(\frac{(S_i p^*)_{\hat{X}_i^m}}{(S_i p')_{\hat{X}_i^m}}\right),$$
is the maximum of the sum of positive-mean random variables, and thus converges to is average (c.f., Lemma 10.5 in [24]). Namely,
$$\lim_{N\to\infty}\max_{1\leq n\leq N}\frac{\widehat{\mathrm{KL}}_i(n)}{N} \to D(S_i p^*\|S_i p')$$
almost surely. Therefore,
$$\lim_{T\to\infty}\frac{\max_{\{N_i(T)\}\in\mathbb{N}^N,\sum_{i\in[N]}N_i(T)D(S_i p^*\|S_i p')<(1-\epsilon)\log T}\sum_{i\in[N]}\widehat{\mathrm{KL}}_i(N_i(T))}{\log T} \to 1-\epsilon$$
almost surely. By using this fact and $1-\epsilon/2 > 1-\epsilon$, we have
$$\Pr\left[\max_{\{N_i(T)\}\in\mathbb{N}^N,\sum_{i\in[N]}N_i(T)D(S_i p^*\|S_i p')<(1-\epsilon)\log T}\sum_{i\in[N]}\widehat{\mathrm{KL}}_i(N_i(T)) > \left(1-\frac{\epsilon}{2}\right)\log T\right] = \mathrm{o}(1).$$
In summary, we obtain $\Pr\left[\mathcal{D}_{1\backslash 2}\right] = \mathrm{o}(1)$.

**Last step:** we here have
$$\begin{aligned}
\mathcal{D}_1 &= \left\{\sum_{i\in[N]} N_i(T)D(S_i p^*\|S_i p') < (1-\epsilon)\log T\right\} \cap \left\{N_{i'}(T) < \sqrt{T}\right\}\\
&\supseteq \left\{\sum_{i\in[N]} N_i(T)D(S_i p^*\|S_i p') + \frac{(1-\epsilon)\log T}{\sqrt{T}}N_{i'}(T) < (1-\epsilon)\log T\right\},
\end{aligned}$$
where we used the fact that $\{A < C\} \cap \{B < C\} \supseteq \{A+B < C\}$ for $A, B > 0$ in the last line. Note that, by using the result of the previous steps, $\Pr[\mathcal{D}_1] = \Pr[\mathcal{D}_{12}] + \Pr[\mathcal{D}_{1\backslash 2}] = \mathrm{o}(1)$. By using the complementary of this fact,
$$\Pr\left[\sum_{i\in[N]} N_i(T)D(S_i p^*\|S_i p') + \frac{(1-\epsilon)\log T}{\sqrt{T}}N_{i'}(T) \geq (1-\epsilon)\log T\right] \geq \Pr[\mathcal{D}_1^c] = 1 - \mathrm{o}(1).$$

Using the Markov inequality yields

$$\mathbb{E}\left[\sum_{i\in[N]} N_i(T)D(S_ip^*\|S_ip') + \frac{(1-\epsilon)\log T}{\sqrt{T}}N_{i'}(T)\right] \geq (1-\epsilon)(1-\mathrm{o}(1))\log T. \qquad (14)$$

Because $\mathbb{E}[N_{i'}(T)]$ is subpolynomial as a function of $T$ due to the consistency, the second term in LHS of (14) is $\mathrm{o}(1)$ and thus negligible. Lemma 1 follows from the fact that (14) holds for sufficiently small $\epsilon$ and arbitrary $p' \in \mathrm{int}(\mathcal{C}_1^c)$. $\qquad\square$

*Proof of Theorem 2.* Assume that there exists $\delta > 0$ and a sequence $T_1 < T_2 < T_3 < \cdots$ such that for all $t$

$$\mathbb{E}[\mathrm{Regret}(T_t)] < (1-\delta)C_1^*(p^*, \{p_i^*\})\log T_t \,,$$

that is,

$$\sum_{i\neq 1} \frac{\mathbb{E}[N_i(T_t)]}{(1-\delta)\log T_t}(L_i - L_1)^\top p^* < C_1^*(p^*, \{p_i^*\})\,.$$

From the definition of $C_1^*$, there exists $q_t' \in \{q \in \mathrm{cl}(\mathcal{C}_1^c) : p_1^* = S_jq\} =: \mathcal{S}$ such that

$$\sum_{i\neq 1} \frac{\mathbb{E}[N_i(T_t)]}{(1-\delta)\log T_t}D(p_i^*\|S_iq_t') < 1\,.$$

Since $\mathcal{S}$ is compact, there exists a subsequence $t_0 < t_1 < \cdots$ such that $\lim_{u\to\infty} q_{t_u}' = q'$ for some $q' \in \mathcal{S}$. Therefore from the lower semicontinuity of the divergence we obtain

$$1 \geq \sum_{i\neq 1} \liminf_{u\to\infty} \frac{\mathbb{E}[N_i(T_t)]}{(1-\delta)\log T_t}D(p_i\|S_iq_{t_u}')$$

$$\geq \sum_{i\neq 1} \liminf_{t\to\infty} \frac{\mathbb{E}[N_i(T_t)]}{(1-\delta)\log T_t}D(p_i\|S_iq')$$

$$= \sum_i \liminf_{t\to\infty} \frac{\mathbb{E}[N_i(T_t)]}{(1-\delta)\log T_t}D(p_i\|S_iq')\,,$$

which contradicts Lemma 1.

$\qquad\square$

## C   The $N$-armed Bandit Problem as Partial Monitoring

In Section 6, we have introduced PM-DMED-Hinge, an asymptotically optimal algorithm for partial monitoring. In this appendix, we prove that this algorithm also has an optimal regret bound of the $N$-armed bandit problem when we run it on an $N$-armed bandit game represented as an instance of partial monitoring.

In the $N$-armed bandit problem, the learner selects one of $N$ actions (arms) and receives a corresponding reward. This problem can be considered as a special case of partial monitoring in which the learner directly observes the loss matrix. For example, three-armed Bernoulli bandit can be represented by the following loss and feedback matrices, and the strategy:

$$L = H = \begin{pmatrix} 2 & 1 & 2 & 1 & 2 & 1 & 2 & 1 \\ 2 & 2 & 1 & 1 & 2 & 2 & 1 & 1 \\ 2 & 2 & 2 & 2 & 1 & 1 & 1 & 1 \end{pmatrix}, \text{ and } p^* = \begin{pmatrix} (1-\mu_1)(1-\mu_2)(1-\mu_3) \\ \mu_1(1-\mu_2)(1-\mu_3) \\ (1-\mu_1)\mu_2(1-\mu_3) \\ \mu_1\mu_2(1-\mu_3) \\ (1-\mu_1)(1-\mu_2)\mu_3 \\ \mu_1(1-\mu_2)\mu_3 \\ (1-\mu_1)\mu_2\mu_3 \\ \mu_1\mu_2\mu_3 \end{pmatrix}, \quad (15)$$

where $\mu_1, \mu_2,$ and $\mu_3$ are the expected rewards of the actions. Signals 1 and 2 correspond to the rewards of 1 and 0 generated by the selected arm, respectively. More generally, $N$-armed Bernoulli

bandit is represented as an instance of partial monitoring in which the loss and feedback matrices are the same $N \times 2^N$ matrix

$$l_{i,j} = h_{i,j} = \mathbb{1}[(j - 1 \bmod 2^i) < 2^{i-1}] + 1,$$

where mod denotes the modulo operation. This problem is associated with $N$ parameters $\mu_1, \mu_2, \ldots, \mu_N$ that correspond to the expected rewards of the actions. For the ease of analysis, we assume $\{\mu_i\}$ are in $(0, 1)$ and different from each other. Without loss of generality, we assume $1 > \mu_1 > \mu_2 > \cdots > \mu_N > 0$, and thus action 1 is the optimal action. The opponent's strategy is

$$p_j^* = \prod_{i \in [N]} \left( \mu_i + (1 - 2\mu_i)\mathbb{1}[(j - 1 \bmod 2^i) < 2^{i-1}] \right).$$

Note that $\mu_i = (S_i p^*)_1$.

*Proof of Corollary 4.* In the following, we prove that the regularity conditions in Theorem 3 are always satisfied in the case of the $N$-armed bandit. During the proof we also show that $C_1^*(p^*, \{p_i^*\})$ is equal to the optimal constant factor of Lai and Robbins [2].

Because signal 1 corresponds to the reward of 1, we can define $\hat{\mu}_i(q) = (S_i q)_1$, and thus

$$\mathcal{C}_i = \{q \in \mathcal{P}_M : \forall_{i' \neq i} \, \hat{\mu}_i(q) \geq \hat{\mu}_{i'}(q)\}.$$

First, we show the uniqueness of $\mathcal{R}_1^*(p, \{p_i, \delta_i\})$ at $p = p^*$, $\{p_i\} = S_i p^*$, $\delta_i = 0$. It is easy to check

$$D(p_i^* \| S_i q) = d(\hat{\mu}_i(p^*) \| \hat{\mu}_i(q)) = d(\mu_i \| \hat{\mu}_i(q)),$$

where $d(a\|b)$ is the KL divergence between two Bernoulli distributions with parameters $a$ and $b$. Then

$$\mathcal{R}_1(\{p_i^*\}) = \left\{ \{r_i\}_{i \neq 1} \in [0, \infty)^{N-1} : \inf_{q \in \mathrm{cl}(\mathcal{C}_1^c) : p_i^* = S_1 q} \sum_i r_i D(p_i^* \| S_i q) \geq 1 \right\}$$

$$= \left\{ \{r_i\}_{i \neq 1} \in [0, \infty)^{N-1} : \inf_{q \in \mathrm{cl}(\mathcal{C}_1^c) : \mu_1 = \hat{\mu}_1(q)} \sum_i r_i D(p_i^* \| S_i q) \geq 1 \right\}$$

$$= \left\{ \{r_i\}_{i \neq 1} : r_i \geq \frac{1}{d(\mu_i \| \mu_1)} \right\}, \tag{16}$$

where the last inequality follows from the fact that

$$\{q \in \mathrm{cl}(\mathcal{C}_1^c) : \hat{\mu}_1(q) = \mu_1\} = \{q \in \mathcal{P}_M : \hat{\mu}_1(q) = \mu_1, \exists_{i \neq 1} \hat{\mu}_i(q) \geq \mu_1\}.$$

By Eq. (16), the regret minimizing solution is

$$C_1^*(p^*, \{p_i^*\}) = \sum_{i \neq 1} \frac{\Delta_i}{d(\mu_i \| \mu_1)},$$

and

$$\mathcal{R}_1^*(p^*, \{p_i^*\}) = \left\{ \{r_i\}_{i \neq 1} : r_i = \frac{1}{d(\mu_i \| \mu_1)} \right\},$$

which is unique.

Second, we show that $\mathrm{cl}(\mathrm{int}(\mathcal{C}_1^c) \cap \mathcal{S}_\delta) = \mathrm{cl}(\mathrm{cl}(\mathcal{C}_1^c) \cap \mathcal{S}_\delta)$ for sufficiently small $\delta \geq 0$. Note that,

$$\mathrm{cl}(\mathcal{C}_1^c) \cap \mathcal{S}_\delta = \{q \in \mathcal{P}_M : \exists_{i' \neq 1} \hat{\mu}_1(q) \leq \hat{\mu}_{i'}(q), d(\mu_1 \| \hat{\mu}_1(q)) \leq \delta\}$$

and

$$\mathrm{int}(\mathcal{C}_1^c) \cap \mathcal{S}_\delta = \{q \in \mathcal{P}_M : \exists_{i' \neq 1} \hat{\mu}_1(q) < \hat{\mu}_{i'}(q), d(\mu_1 \| \hat{\mu}_1(q)) \leq \delta\}.$$

To prove

$$\mathrm{cl}(\mathcal{C}_1^c) \cap \mathcal{S}_\delta \subset \mathrm{cl}(\mathrm{int}(\mathcal{C}_1^c) \cap \mathcal{S}_\delta), \tag{17}$$

it suffices to show that, an open ball centered at any position in

$$\{q \in \mathcal{P}_M : \exists_{i' \neq 1} \hat{\mu}_1(q) = \hat{\mu}_{i'}(q), d(\mu_1 \| \hat{\mu}_1(q)) \leq \delta\} \supset (\mathrm{cl}(\mathcal{C}_1^c) \cap \mathcal{S}_\delta) \setminus (\mathrm{int}(\mathcal{C}_1^c) \cap \mathcal{S}_\delta)$$

Figure 4: Regret-round semilog plots of algorithms. The regrets are averaged over 100 runs. LB-Theory is the asymptotic regret lower bound of Lai and Robbins [2].

contains a point in $\text{int}(\mathcal{C}_1^c) \cap \mathcal{S}_\delta$. This holds because we can make a slight move towards the direction of increasing $\hat{\mu}_{i'}$: we can always find $q'$ in an open ball centered at $q$ such that $\hat{\mu}_{i'}(q') > \hat{\mu}_{i'}(q)$ and $\hat{\mu}_1(q') = \hat{\mu}_1(q)$ because of (i) the fact that there always exists $q \in \mathcal{P}_M$ such that $\{q \in \mathcal{P}_M, \forall_{i \in [N]} \hat{\mu}_i(q) = \mu_i\}$ for arbitrary $\{\mu_i\} \in (0, 1)^N$ and (ii) the continuity of the $\hat{\mu}_i$ operator. Therefore, any open ball centered at $q \in \text{cl}(\mathcal{C}_1^c) \cap \mathcal{S}_\delta$ contains an element of $\text{int}(\mathcal{C}_1^c) \cap \mathcal{S}_\delta$, by which we obtain (17). By using (17), we have

$$\text{cl}(\text{cl}(\mathcal{C}_1^c) \cap \mathcal{S}_\delta) \subset \text{cl}(\text{cl}(\text{int}(\mathcal{C}_1^c) \cap \mathcal{S}_\delta)) = \text{cl}(\text{int}(\mathcal{C}_1^c) \cap \mathcal{S}_\delta), \tag{18}$$

where we used the fact that $\text{cl}(\text{cl}(X)) = \text{cl}(X)$. Combining (18) with the fact that $\text{cl}(\text{cl}(\mathcal{C}_1^c) \cap \mathcal{S}_\delta) \supset \text{cl}(\text{int}(\mathcal{C}_1^c) \cap \mathcal{S}_\delta)$ yields $\text{cl}(\text{cl}(\mathcal{C}_1^c) \cap \mathcal{S}_\delta) = \text{cl}(\text{int}(\mathcal{C}_1^c) \cap \mathcal{S}_\delta)$.

Therefore, in the $N$-armed Bernoulli bandit problem, the regularity conditions are always satisfied and $C_1^*(p^*, \{p_i^*\})$ matches the optimal coefficient of the logarithmic regret bound. From Theorem 3, if we run PM-DMED-Hinge in this game, its expected regret is asymptotically optimal in view of the $N$-armed bandit problem. □

## C.1 Experiment

We also assessed the performance of PM-DMED and other algorithms in solving the three-armed Bernoulli bandit game defined by (15) with parameters $\mu_1 = 0.4$, $\mu_2 = 0.3$, and $\mu_3 = 0.2$. The settings of the algorithms are the same as that of the main paper. The results of simulations are shown in Figure 4. LB-Theory is the regret lower bound of Lai and Robbins [2], that is, $\sum_{i \neq 1} \frac{\Delta_i \log t}{d(\mu_i \| \mu_1)}$. The slope of PM-DMED quickly approaches that of LB-Theory, which is empirical evidence that PM-DMED has optimal performance in $N$-armed bandits.

## D   Optimality of PM-DMED-Hinge

In this appendix we prove Theorem 3. First we define distances among distributions. For distributions $p_i, p_i' \in \mathcal{P}_A$ of symbols we use the total variation distance

$$\|p_i - p_i'\| = \frac{1}{2} \sum_{a=1}^{A} |(p_i)_a - (p_i')_a|.$$

For distributions $p, p' \in \mathcal{P}_M$ of outcomes, we identify $p$ with the set $\{p' : \forall i, S_i p = S_i p'\}$ and define

$$\|p - p'\|_M = \max_i \|S_i p - S_i p'\|.$$

For $\mathcal{Q} \subset \mathcal{P}_M$ we define

$$\|p - \mathcal{Q}\|_M = \inf_{p' \in \mathcal{Q}} \|p - p'\|_M.$$

In the following, we use Pinsker's inequality given below many times.

$$D(p_i \| q_i) \geq 2\|p_i - q_i\|^2 \,.$$

Let

$$\rho_{i,L} = \sup_{\lambda > 0} \frac{1}{\lambda} \min_{x \in \mathcal{C}_{i,\lambda}} \|x - \mathcal{C}_i^c\|_M \,,$$

$$\nu_{i,L} = \sup_{\lambda > 0} \frac{1}{\lambda} \max_{x \in \mathcal{C}_{i,\lambda}^c} \|x - \mathcal{C}_i^c\|_M \,.$$

Note that these two constants are positive from the global observability.

### D.1 Properties of regret lower bound

In this section, we give Lemma 5 and Theorem 6 that are about the functions $C_j^*(p, \{p_i, \delta_i\})$ and $\mathcal{R}_j^*(p, \{p_i, \delta_i\})$. In the following, we always consider these functions on $p \in \mathcal{P}_M$, $p_i \in \{S_i p : \mathrm{supp}(p) \subset \mathrm{supp}(p^*)\}$ and $\delta_i \geq 0$, where $\mathrm{supp}(\cdot)$ denotes the support of the distribution.

We define

$$L_{\max} = \max_{i',j'} l_{i',j'} \,.$$

**Lemma 5.** *Let $p \in \mathcal{C}_{j,\alpha}$ and $\{p_i, \delta_i\}$ be satisfying $\|p_i - S_i p\| \leq \alpha \rho_{j,L}/2$ and $\delta_i \leq (\alpha \rho_{j,L})^2/4$ for all $i$. Then*

$$C_j^*(p, \{p_i, \delta_i\}) \leq \frac{4NL_{\max}}{(\alpha \rho_{j,L})^2} \,. \tag{19}$$

*Furtheremore, $\mathcal{R}_j(\{p_i, \delta_i\})$ is nonempty and*

$$\mathcal{R}_j^*(p, \{p_i, \delta_i\}) \subset \left[0, \frac{4NL_{\max}}{(\rho_{j,L})^2 \alpha^3}\right]^{N-1} \,.$$

*Proof of Lemma 5.* Since $\|p - \mathcal{C}_1^c\|_M \geq \alpha \rho_{j,L}$, there exists $i = i(q)$ for any $q \in \mathcal{C}_1^c$ such that

$$\|S_i q - S_i p\| \geq \alpha \rho_{j,L} \,.$$

For this $i$ we have

$$
\begin{aligned}
D(p_i \| S_i q) - \delta_i &\geq 2\|p_i - S_i q\|^2 - \delta_i \\
&\geq 2(\|S_i q - S_i p\| - \|p_i - S_i p\|)_+^2 - \delta_i \\
&\geq (\alpha \rho_{j,L})^2/2 - \delta_i \\
&\geq (\alpha \rho_{j,L})^2/4 \,.
\end{aligned}
$$

Thus, by letting $r_i = 4/(\alpha \rho_{j,L} \alpha)^2$ for all $i \neq j$ we have

$$\{r_i\}_{i \neq j} \in \mathcal{R}_j(\{p_i, \delta_i\}) \,,$$

which implies (19). On the other hand it holds for any $\{r_i^*\}_{i \neq j} \in \mathcal{R}_j^*(p, \{p_i, \delta_i\})$ from $p \in \mathcal{C}_{j,\alpha}$ that

$$C_j^*(p, \{p_i, \delta_i\}) = \sum_{i \neq j} r_i^* L_i^\top p \geq \max_{i \neq j} r_i^* \alpha$$

and therefore we have

$$\max_{i \neq j} r_i^* \leq \frac{4NL_{\max}}{(\rho_{j,L})^2 \alpha^3} \,.$$

$\square$

**Theorem 6.** *Assume that the regularity conditions in Theorem 3 hold. Then the point-to-set map $\mathcal{R}_1^*(p, \{p_i, \delta_i\})$ is (i) nonempty near $p = p^*, p_i = S_i p^*, \delta_i = 0$ and (ii) continuous at $p = p^*, p_i = S_i p^*, \delta_i = 0$.*

See Hogan [25] for definitions of terms such as continuity of point-to-set maps.

*Proof of Theorem 6.* Define

$$
\bar{\mathcal{R}}_1(\{p_i, \delta_i\}) = \left\{ \{r_i\}_{i \neq 1} \in [0, \xi]^{N-1} : \inf_{q \in \mathrm{cl}(\mathcal{C}_1^c): D(p_1 \| S_1 q) \leq \delta_1} \sum_{i \neq 1} r_i (D(p_i \| S_i q) - \delta_i)_+ \geq 1 \right\}
$$

for

$$
\xi = \frac{4 N L_{\max}}{(\rho_{1,L})^2 (\max_{i \neq 1} L_i^\top p^* - L_1^\top p^*)^3} .
$$

Note that $p^* \in \mathcal{C}_{1,\alpha}$ for $\alpha \leq \max_{i \neq 1} L_i^\top p^* - L_1^\top p^*$. From Lemma 5, near $p = p^*, p_i = S_i p^*, \delta_i = 0$,

$$
\bar{\mathcal{R}}_1(\{p_i, \delta_i\}) \supset \mathcal{R}_1^*(p, \{p_i, \delta_i\})
$$

and

$$
C_1^*(p, \{p_i, \delta_i\}) = \inf_{\{r_i\}_{i \neq 1} \in \bar{\mathcal{R}}_1(\{p_i, \delta_i\})} \sum_{i \neq 1} r_i (L_i - L_1)^\top p
$$

hold. Since the function

$$
\sum_i r_i (D(p_i \| S_i q) - \delta_i)_+
$$

is continuous in $\{r_i\}$, $\bar{\mathcal{R}}_1(\{p_i, \delta_i\})$ is a closed set and therefore $\mathcal{R}_1^*(p, \{p_i, \delta_i\})$ is nonempty near $p = p^*, p_i = S_i p^*, \delta_i = 0$.

From the continuity of $D(p_i \| S_i q)$ at any $q$ such that $D(p_i \| S_i q) < \infty$, we have

$$
\inf_{q \in \mathrm{cl}(\mathcal{C}_1^c) \cap \mathcal{S}_{\delta_1}} \sum_{i \neq 1} r_i (D(p_i \| S_i q) - \delta_i)_+ = \inf_{q \in \mathrm{cl}(\mathrm{cl}(\mathcal{C}_1^c) \cap \mathcal{S}_{\delta_1})} \sum_{i \neq 1} r_i (D(p_i \| S_i q) - \delta_i)_+
$$

$$
= \inf_{q \in \mathrm{cl}(\mathrm{int}(\mathcal{C}_1^c) \cap \mathcal{S}_{\delta_1})} \sum_{i \neq 1} r_i (D(p_i \| S_i q) - \delta_i)_+
$$

$$
= \inf_{q \in \mathrm{int}(\mathcal{C}_1^c) \cap \mathcal{S}_{\delta_1}} \sum_{i \neq 1} r_i (D(p_i \| S_i q) - \delta_i)_+ .
$$

Thus, we have

$$
\bar{\mathcal{R}}_1(\{p_i, \delta_i\}) = \left\{ \{r_i\}_{i \neq 1} \in [0, \xi]^{N-1} : \inf_{q \in \mathrm{int}(\mathcal{C}_1^c): D(p_1 \| S_1 q) \leq \delta_1} \sum_i r_i (D(p_i \| S_i q) - \delta_i)_+ \geq 1 \right\} . \tag{20}
$$

Since the objective function $\sum_{i \neq j} r_i (L_i - L_j)^\top p$ is continuous in $\{r_i\}$ and $p$, and (20) is compact, now it suffices to show that (20) is continuous in $\{p_i, \delta_i\}$ at $\{S_i p^*, 0\}$ to prove the theorem from [25, Corollary 8.1].

First we show that $\bar{\mathcal{R}}_1(\{p_i, \delta_i\})$ is closed at $\{S_i p^*, 0\}$. Consider $\{r_i^{(m)}\}_{i \neq 1} \in \bar{\mathcal{R}}_1(\{p_i^{(m)}, \delta_i^{(m)}\})$ for a sequence $\{p_i^{(m)}, \delta_i^{(m)}\}_i$ which converges to $\{S_i p^*, 0\}_i$ as $m \to \infty$. We show that $\{r_i\}_{i \neq 1} \in \bar{\mathcal{R}}_1(\{S_i p^*, 0\})$ if $r_i^{(m)} \to r_i$ as $m \to \infty$.

Take an arbitrary $q \in \mathrm{int}(\mathcal{C}_1^c)$ such that $D(S_1 p^* \| S_1 q) = 0$. Since $\|S_1 p^* - p_1^{(m)}\| \to 0$ and $p_1 \in \{S_1 p : \mathrm{supp}(p) \subset \mathrm{supp}(p^*)\}$, there exists $\tilde{p}^{(m)}$ such that $p_1^{(m)} = S_1 \tilde{p}^{(m)}$ and $\|p^* - \tilde{p}^{(m)}\|_M \to 0$.

Thus, from $q \in \text{int}(\mathcal{C}_1^c)$, it holds for sufficiently large $m$ that $q^{(m)} = q - p^* + \tilde{p}^{(m)} \in \text{int}(\mathcal{C}_1^c)$. For this $q^{(m)}$ we have

$$D(p_1^{(m)} \| S_1 q^{(m)}) \leq D(S_1 \tilde{p}^{(m)} \| S_1 (q - p^* + \tilde{p}^{(m)})) = 0 \leq \delta_1 \,.$$

That is, $q^{(m)} \in \text{int}(\mathcal{C}_1^c) \cap \mathcal{S}_{\delta_1}$. Therefore, for sufficiently large $m$ we have

$$\sum_i r_i^{(m)} (D(p_i \| S_i q^{(m)}) - \delta_i^{(m)})_+ \geq 0$$

and, letting $m \to \infty$,

$$\sum_i r_i D(p_i \| S_i q) \geq 0 \,.$$

This means that $\{r_i\}_{i \neq 1} \in \bar{\mathcal{R}}_1(\{p_i, \delta_i\})$, that is, $\bar{\mathcal{R}}_1(\{p_i, \delta_i\})$ is closed at $\{S_i p^*, 0\}$.

Next we show that $\bar{\mathcal{R}}_1(\{p_i, \delta_i\})$ is open at $\{S_i p^*, 0\}$. Consider $\{r_i\}_{i \neq 1} \in \bar{\mathcal{R}}_1(\{S_i p^*, 0\})$ and a sequence $\{p_i^{(m)}, \delta_i^{(m)}\}_i$ which converges to $\{S_i p^*, 0\}_i$ as $m \to \infty$. We show that there exists a sequence $\{r_i^{(m)}\}_{i \neq 1} \in \bar{\mathcal{R}}_1(\{p_i^{(m)}, \delta_i^{(m)}\})$ such that $r_i^{(m)} \to r_i$.

Consider the optimal value function

$$v(\{p_i^{(m)}, \delta_i^{(m)}\}) = \inf_{q \in \text{cl}(\mathcal{C}_1^c) \cap \mathcal{S}_{\delta_1}} \sum_i r_i (D(p_i^{(m)} \| S_i q) - \delta_i^{(m)})_+ \,. \tag{21}$$

Since the feasible region of (21) is closed at $p_i = S_i p^*$, $\delta_i = 0$ and the objective function of (21) is lower semicontinuous in $q$, $\{p_i, \delta_i\}$ we see that $v(\{p_i^{(m)}, \delta_i^{(m)}\})$ is lower semicontinuous from [25, Theorem 2]. Therefore, for any $\epsilon > 0$ there exists $m_0 > 0$ such that for all $m \geq m_0$

$$v(\{p_i^{(m)}, \delta_i^{(m)}\}) \geq (1 - \epsilon) v(\{S_i p^*, 0\}) \geq 1$$

since $v(\{S_i p^*, 0\}) \geq 1$ from $r_i \in \bar{\mathcal{R}}_1(\{S_i p^*, 0\})$. Thus, by letting $r_i^{(m)} := r_i / (1 - \epsilon)$ we have

$$\inf_{v \in \text{cl}(\mathcal{C}_1^c) \cap \mathcal{S}_{\delta_1}} \sum_i r_i^{(m)} (D(p_i^{(m)} \| S_i q^{(m)}) - \delta_i^{(m)})_+ \geq 1 \,,$$

that is, $\{r_i^{(m)}\}_{i \neq 1} \in \bar{\mathcal{R}}_1(\{p_i^{(m)}, \delta_i^{(m)}\})$. $\qquad \square$

### D.2 Regret analysis of PM-DMED-Hinge

Let $\hat{p}_{i,n} \in [0,1]^A$ be the empirical distribution of the symbols from the action $i$ when the action $i$ is selected $n$ times. Then we have $\hat{p}_i(t) = \hat{p}_{i,N_i(t)}$. Let $P_{i,n_i}(u) = \Pr[D(\hat{p}_{i,n_i} \| S_i p^*) \geq u]$. Then, from the large deviation bound on discrete distributions (Theorem 11.2.1 in Cover and Thomas [26]), we have

$$P_{i,n_i}(u) \leq (n_i + 1)^A \mathrm{e}^{-n_i u} \,. \tag{22}$$

We also define

$$\mathcal{H}(\{p_i, n_i\}) = \{i \subset [N] : D(p_i \| S_i p^*) - f(n_i) > 0\}.$$

For

$$0 < \delta \leq \|p^* - \mathcal{C}_1^c\|_M^2 / 8 \tag{23}$$

define events

$$\mathcal{A}(t) = \{\hat{p}(t) \in \mathcal{C}_1\}$$
$$\mathcal{A}'(t) = \{\hat{p}(t) \in \mathcal{C}_{1,\alpha(t)}\}$$
$$\mathcal{B}(t) = \bigcap_i \{\|\hat{p}_i(t) - S_i p^*\| \leq \sqrt{\delta}\}$$
$$\mathcal{C}(t) = \{\hat{i}(t) \notin \mathcal{H}(\{\hat{p}_i(t), N_i(t)\}), \mathcal{H}(\{\hat{p}_i(t), N_i(t)\}) \neq \emptyset\}$$
$$= \left\{ D(\hat{p}_{\hat{i}(t)}(t) \| S_{\hat{i}(t)} p^*) \leq f(N_{\hat{i}(t)}(t)), \bigcup_i \{D(\hat{p}_i(t) \| S_i p^*) > f(N_i(t))\} \right\}$$
$$\mathcal{D}(t) = \bigcap_i \{D(\hat{p}_i(t) \| S_i \hat{p}(t)) \leq f(N_i(t))\}$$
$$\mathcal{E}(t) = \left\{ \max_i f(N_i(t)) \leq \min \left\{ 2\delta, (\rho_{1,L}\alpha(t))^2/4 \right\}, \right. \tag{24}$$
$$\left. \min_i N_i(t) \geq \max\{c\sqrt{\log t}, (\log\log T)^{1/3}\}, 2\nu_{1,L}\alpha(t) \leq \|p^* - \mathcal{C}_1^c\|_M \right\},$$

where we write $\{\mathcal{T}, \mathcal{U}\}$ instead of $\{\mathcal{T} \cap \mathcal{U}\}$ for events $\mathcal{T}$ and $\mathcal{U}$.

*Proof of theorem 3.* Since $\mathcal{A}'(t) \subset \mathcal{A}(t)$, the whole sample space is covered by

$$\{\mathcal{A}'(t), \mathcal{B}(t)\} \cup \{\mathcal{A}'(t), \mathcal{B}^c(t)\} \cup \{\mathcal{A}(t), (\mathcal{A}'(t))^c\} \cup \{\mathcal{A}^c(t), \mathcal{C}(t)\} \cup \{\mathcal{A}^c(t), \mathcal{C}^c(t)\}$$
$$\subset \{\mathcal{A}'(t), \mathcal{B}(t), \mathcal{D}(t), \mathcal{E}(t)\} \cup \{\mathcal{A}'(t), \mathcal{B}^c(t), \mathcal{D}(t), \mathcal{E}(t)\} \cup \{\mathcal{A}(t), (\mathcal{A}'(t))^c, \mathcal{D}(t), \mathcal{E}(t)\} \cup \{\mathcal{A}^c(t), \mathcal{C}(t)\}$$
$$\cup \{\mathcal{A}^c(t), \mathcal{C}^c(t), \mathcal{D}(t), \mathcal{E}(t)\} \cup \mathcal{D}^c(t) \cup \mathcal{E}^c(t). \tag{25}$$

Let $J_i(t)$ denote the event that action $i$ is newly added into the list $L_N$ at the $t$-th round and $J'_i(t) \subset J_i(t)$ denote the event that $J_i(t)$ occurred by Step 6 of Algorithm 3. Note that if $\{\mathcal{A}'(t), \mathcal{D}(t), \mathcal{E}(t)\}$ occurred then $J_i(t)$ is equivalent to $J'_i(t)$. Combining this fact with (25) we can bound the regret as

$$\text{Regret}(T) \leq \sum_{i \neq 1} \Delta_i \sum_{t=1}^{T} \mathbb{1}[J_i(t)] + N$$
$$\leq \sum_{i \neq 1} \Delta_i \sum_{t=1}^{T} \left( \mathbb{1}[J'_i(t), \mathcal{A}'(t), \mathcal{B}(t), \mathcal{D}(t), \mathcal{E}(t)] + \mathbb{1}[J'_i(t), \mathcal{A}(t), \mathcal{B}^c(t), \mathcal{D}(t), \mathcal{E}(t)] \right.$$
$$+ \mathbb{1}[J_i(t), \mathcal{A}(t), (\mathcal{A}'(t))^c, \mathcal{D}(t), \mathcal{E}(t)] + \mathbb{1}[J_i(t), \mathcal{A}^c(t), \mathcal{C}^c(t), \mathcal{D}(t), \mathcal{E}(t)]$$
$$+ \left. \mathbb{1}[J_i(t), \mathcal{D}^c(t) \cup \mathcal{E}^c(t)] \right) + \left( \sum_{i \neq 1} \Delta_i \right) \sum_{t=1}^{T} \mathbb{1}[\mathcal{A}^c(t), \mathcal{C}(t)] + N.$$

The following Lemmas 7–13 bound the expectation of each term and complete the proof. $\square$

**Lemma 7.** *Let $\{r_i^*\}_{i \neq 1}$ be the unique member of $\mathcal{R}_j^*(p^*, \{S_i p^*, 0\})$. Then there exists $\epsilon_\delta > 0$ such that $\lim_{\delta \to 0} \epsilon_\delta = 0$ and for all $i \neq 1$*

$$\sum_{t=1}^{T} \mathbb{1}[J'_i(t), \mathcal{A}'(t), \mathcal{B}(t), \mathcal{D}(t), \mathcal{E}(t)] \leq (1 + \epsilon_\delta) r_i^* \log T + 1.$$

**Lemma 8.**

$$\mathbb{E}\left[ \sum_{t=1}^{T} \mathbb{1}[J'_i(t), \mathcal{A}'(t), \mathcal{B}^c(t), \mathcal{D}(t), \mathcal{E}(t)] \right] = o(\log T).$$

**Lemma 9.**

$$\mathbb{E}\left[ \sum_{t=1}^{T} \mathbb{1}[\mathcal{A}^c(t), \mathcal{C}(t)] \right] = O(1).$$

**Lemma 10.**

$$\mathbb{E}\left[\sum_{t=1}^{T} \mathbb{1}\left[J_i(t),\, \mathcal{A}(t),\, (\mathcal{A}'(t))^c,\, \mathcal{D}(t),\, \mathcal{E}(t)\right]\right] = \mathrm{O}(1)\,.$$

**Lemma 11.**

$$\mathbb{E}\left[\sum_{t=1}^{T} \mathbb{1}\left[J_i(t),\, \mathcal{A}^c(t),\, \mathcal{C}^c(t),\, \mathcal{D}(t),\, \mathcal{E}(t)\right]\right] = \mathrm{O}(1)\,.$$

**Lemma 12.**

$$\mathbb{E}\left[\sum_{t=1}^{T} \mathbb{1}\left[J_i(t),\, \mathcal{D}^c(t)\right]\right] = \mathrm{O}(1)\,.$$

**Lemma 13.**

$$\sum_{t=1}^{T} \mathbb{1}\left[J_i(t),\, \mathcal{E}^c(t)\right] = \mathrm{o}(\log T)\,.$$

*Proof of Lemma 7.* From $\mathcal{D}(t)$ we have

$$\sum_i N_i(t)(D(\hat{p}_i(t)\|S_i\hat{p}(t)) - f(N_i(t)))_+ = 0\,. \tag{26}$$

Here assume that $\|\hat{p}(t) - p^*\|_M > 2\sqrt{\delta}$. Then

$$
\begin{aligned}
\max_i D(\hat{p}_i(t)\|S_i\hat{p}(t)) &\geq 2\max_i \|\hat{p}_i(t) - S_i\hat{p}(t)\|^2 \quad \text{(by Pinsker's inequality)} \\
&\geq 2\max_i(\|S_ip^* - S_i\hat{p}(t)\| - \|S_ip^* - \hat{p}_i(t)\|)_+^2 \\
&\geq 2\max_i(\|S_ip^* - S_i\hat{p}(t)\| - \sqrt{\delta})_+^2 \quad \text{(by definition of } \mathcal{B}(t)) \\
&> 2\delta \\
&\geq f(N_i(t))\,, \quad \text{(by definition of } \mathcal{E}(t))
\end{aligned}
$$

which contradicts (26) and we obtain $\|\hat{p}(t) - p^*\|_M \leq 2\sqrt{\delta}$. Furthermore, from $\mathcal{B}(t)$ and $\mathcal{E}(t)$ we have

$$\bigcap_i \{\|\hat{p}_i(t) - S_ip^*\| \leq \sqrt{\delta}\} \text{ and } \bigcap_i \{f(N_i(t)) \leq 2\delta\}\,,$$

respectively. Since $\mathcal{R}_1^*(p, \{p_i, \delta_i\})$ is continuous at $p = p^*$, $p_i = S_ip^*$, $\delta_i = 0$ from Theorem 6, $r_i \leq (1 + \epsilon_\delta)r_i^*$ for all $\{r_i\}_{i\neq 1} \in \mathcal{R}_{\hat{i}(t)}^*(\hat{p}(t), \{\hat{p}_i(t), f(N_i(t))\})$ where $r_i^*$ is the unique member of $\mathcal{R}_1^*(p^*, \{S_ip^*, 0\})$ and we used the fact that $\mathcal{A}'(t)$ implies $\hat{i}(t) = 1$.

We complete the proof by

$$
\begin{aligned}
&\sum_{t=1}^{T} \mathbb{1}\left[J_i'(t),\, \mathcal{A}'(t),\, \mathcal{B}(t),\, \mathcal{D}(t),\, \mathcal{E}(t)\right] \\
&= \sum_{n=1}^{T} \mathbb{1}\left[\bigcup_{t=1}^{T}\{J_i'(t),\, \mathcal{A}'(t),\, \mathcal{B}(t),\, \mathcal{D}(t),\, \mathcal{E}(t),\, N_i(t) = n\}\right] \\
&\leq \sum_{n=1}^{T} \mathbb{1}\left[\bigcup_{t=1}^{T}\{n/\log t \leq (1 + \epsilon_\delta)r_i^*\}\right] \\
&\leq (1 + \epsilon_\delta)r_i^* \log T + 1\,.
\end{aligned}
$$

$\square$

*Proof of Lemma 8.* First, we obtain from $\mathcal{D}(t)$ and $\mathcal{E}(t)$ that $f(N_i(t)) \leq (\rho_{1,L}\alpha(t))^2/4$ and

$$
\begin{aligned}
\|\hat{p}_i(t) - S_i\hat{p}(t)\| &\leq \sqrt{D(\hat{p}_i(t)\|S_i\hat{p}(t))/2} \\
&\leq \sqrt{f(N_i(t))/2} \\
&\leq \rho_{1,L}\alpha(t)/\sqrt{8}\,.
\end{aligned}
$$

Therefore, from Lemma 5, it holds for any $\{r_i^*\}_{i\neq 1} \in \mathcal{R}_j^*(\hat{p}(t), \{\hat{p}_i(t), f(N_i(t))\})$ that

$$
\begin{aligned}
r_i^* &\leq \frac{4NL_{\max}}{(\rho_{1,L})^2(\alpha(t))^3} \\
&\leq \frac{4NL_{\max}}{(\rho_{1,L})^2(\alpha(T))^3}\,.
\end{aligned}
$$

Now we have

$$
\begin{aligned}
&\mathbb{E}\left[\sum_{t=1}^{\infty} \mathbb{1}\left[J_i'(t),\, \mathcal{A}'(t),\, \mathcal{B}^c(t),\, \mathcal{D}(t),\, \mathcal{E}(t)\right]\right] \\
&\leq \mathbb{E}\left[\sum_{t=1}^{\infty} \mathbb{1}\left[\frac{N_i(t)}{\log T} < \frac{4NL_{\max}}{(\rho_{1,L})^2(\alpha(T))^3},\, \mathcal{B}^c(t),\, \mathcal{E}(t)\right]\right] \\
&\leq \left(\frac{4NL_{\max}\log T}{(\rho_{1,L})^2(\alpha(T))^3} + 1\right)\Pr\left[\bigcup_{t=1}^{T}\{\mathcal{B}^c(t),\, \mathcal{E}(t)\}\right]. \quad (27)
\end{aligned}
$$

Here, note that

$$
\begin{aligned}
\mathcal{B}^c(t) &\subset \bigcup_i \{\|\hat{p}_i(t) - S_ip^*\| \geq \sqrt{\delta}\} \\
&\subset \bigcup_i \{D(\hat{p}_i(t)\|S_ip^*) \geq 2\delta\}\,.
\end{aligned}
$$

Since $N_i(t) \geq (\log\log T)^{1/3}$ holds under event $\mathcal{E}(t)$, we can bound the probability in (27) as

$$
\begin{aligned}
&\Pr\left[\bigcup_{t=1}^{T}\{\mathcal{B}^c(t),\, \mathcal{E}(t)\}\right] \\
&\leq \sum_i \sum_{n_i=(\log\log T)^{1/3}}^{\infty} \Pr[D(\hat{p}_{i,n_i}\|S_ip^*) \geq 2\delta] \\
&\leq N \sum_{n=(\log\log T)^{1/3}}^{\infty} (n+1)^A e^{-2n\delta} \quad \text{(by (22))} \\
&= e^{-\Theta((\log\log T)^{1/3})}
\end{aligned}
$$

and combining this with (27) we have

$$
\begin{aligned}
&\mathbb{E}\left[\sum_{t=1}^{\infty} \mathbb{1}\left[J_i'(t),\, \mathcal{A}'(t),\, \mathcal{B}^c(t),\, \mathcal{D}(t),\, \mathcal{E}(t)\right]\right] \\
&\leq O\left((\log T)(\log\log T)^3\right) e^{-\Theta((\log\log T)^{1/3})} \\
&= o(\log T)\,.
\end{aligned}
$$

$\square$

*Proof of Lemma 9.* Let $\mathcal{G} \in 2^{[N]} \setminus \emptyset$ and $\{n_i\}_{i\in\mathcal{G}} \in \mathbb{N}^{|\mathcal{G}|}$ be arbitrary. Consider the case that

$$
\sum_{i\in\mathcal{G}} n_i(D(\hat{p}_{i,n_i}\|S_ip^*) - f(n_i))_+ < x\,. \quad (28)
$$

for some $x > 0$. Then under events $t \geq \mathrm{e}^x$, $\bigcap_{i \in \mathcal{G}}\{N_i(t) = n_i\}$, $\mathcal{A}^c(t)$, $\mathcal{C}(t)$ and $\mathcal{H}(\{\hat{p}_i(t), f(n_i)\}) = \mathcal{G}$ we have

$$
\min_{p \in \mathcal{C}^c_{\hat{i}(t)} : D(\hat{p}_{\hat{i}(t)}(t) \| S_{\hat{i}(t)} p) \leq f(N_{\hat{i}(t)}(t))} \sum_i N_i(t)(D(\hat{p}_i(t) \| S_i p) - f(N_i(t)))_+
$$

$$
\leq \sum_i n_i(D(\hat{p}_i(t) \| S_i p^*) - f(n_i))_+ < x \leq \log t,
$$

which implies that the condition (9) is satisfied. On the other hand from (10), $\{r_i^*\}$ satisfies

$$
\sum_{i \in \mathcal{G}} (r_i^* \log t)(D(\hat{p}_i(t) \| S_i p) - f(n_i))_+ \geq \log t. \tag{29}
$$

Eqs. (28) and (29) imply that there exists at least one $i \in \mathcal{G}$ such that $r_i^* \log t > N_i(t) = n_i$. This action is selected within $N$ rounds and therefore $N_i(t') = n_i$ never holds for all $t' \geq t + N$. Thus, under the condition (28) it holds that

$$
\sum_t \mathbb{1}\left[\mathcal{A}^c(t), \mathcal{C}(t), \mathcal{H}(\{\hat{p}_i(t), N_i(t)\}) = \mathcal{G}, \bigcap_{i \in \mathcal{G}}\{N_i(t) = n_i\}\right] \leq \mathrm{e}^x + N.
$$

By using this inequality we have

$$
\sum_{t=1}^\infty \mathbb{1}\left[\mathcal{A}^c(t), \mathcal{C}(t)\right]
$$

$$
\leq \sum_{\mathcal{G} \in 2^{[N]} \setminus \emptyset} \sum_{\{n_i\}_{i \in \mathcal{G}} \in \mathbb{N}^{|\mathcal{G}|}} \sum_{t=1}^\infty \mathbb{1}\left[\mathcal{A}^c(t), \mathcal{C}(t), \mathcal{H}(\{\hat{p}_i(t), N_i(t)\}) = \mathcal{G}, \bigcap_{i \in \mathcal{G}}\{N_i(t) = n_i\}\right]
$$

$$
\leq \sum_{\mathcal{G} \in 2^{[N]} \setminus \emptyset} \sum_{\{n_i\}_{i \in \mathcal{G}} \in \mathbb{N}^{|\mathcal{G}|}} \mathbb{1}\left[\bigcap_{i \in \mathcal{G}}\{D(\hat{p}_{i,n_i} \| S_i p^*) \geq f(n_i)\}\right] \left(\exp\left(\sum_{i \in \mathcal{G}} n_i(D(\hat{p}_{i,n_i} \| S_i p^*) - f(n_i))\right) + N\right). \tag{30}
$$

Let
$$
D_i = \sup_n\{\mathrm{ess\,sup}\, D(\hat{p}_{i,n} \| S_i p^*)\} = -\log \min_{j:(S_i p^*)_j > 0}(S_i p^*)_j,
$$

where $(S_i p^*)_j$ is the $j$-th component of $S_i p^*$. Then,

$$
\mathbb{E}\left[\sum_{\{n_i\}_{i \in \mathcal{G}} \in \mathbb{N}^{|\mathcal{G}|}} \mathbb{1}\left[\bigcap_{i \in \mathcal{G}}\{D(\hat{p}_{i,n_i} \| S_i p^*) \geq f(n_i)\}\right] \left(\exp\left(\sum_{i \in \mathcal{G}} n_i(D(\hat{p}_{i,n_i} \| S_i p^*) - f(n_i))\right) + N\right)\right]
$$

$$
\leq \sum_{\{n_i\}_{i \in \mathcal{G}} \in \mathbb{N}^{|\mathcal{G}|}} \left(\prod_{i \in \mathcal{G}} \int_{f(n_i)}^{D_i} \mathrm{e}^{n_i(u_i - f(n_i))} \mathrm{d}(-P_{i,n_i}(u_i)) + N \prod_{i \in \mathcal{G}} (n_i + 1)^A \mathrm{e}^{-n_i f(n_i)}\right).
$$

The first integral is bounded as

$$
\int_{f(n_i)}^{D_i} \mathrm{e}^{n_i(u_i - f(n_i))} \mathrm{d}(-P_i(u_i))
$$

$$
= \left[-\mathrm{e}^{n_i(u_i - f(n_i))} P_i(u_i)\right]_{f(n_i)}^{D_i} + \int_{f(n_i)}^{D_i} n_i \mathrm{e}^{n_i(u_i - f(n_i))} P_i(u_i) \mathrm{d}u_i \quad \text{(integration by parts)}
$$

$$
\leq (n_i + 1)^A \mathrm{e}^{-n_i f(n_i)} + \int_{f(n_i)}^{D_i} n_i(n_i + 1)^A \mathrm{e}^{-n_i f(n_i)} \mathrm{d}u_i
$$

$$
\leq (1 + n_i D_i)(n_i + 1)^A \mathrm{e}^{-n_i f(n_i)}. \tag{31}
$$

Putting (30)–(31) together we have

$$\mathbb{E}\left[\sum_{t=1}^{\infty} \mathbb{1}\left[\mathcal{A}^c(t), \mathcal{C}(t)\right]\right]$$

$$\leq \sum_{\mathcal{G}\in 2^{[N]}\setminus\emptyset} \sum_{\{n_i\}_{i\in\mathcal{G}}\in\mathbb{N}^{|\mathcal{G}|}} \left(\prod_{i\in\mathcal{G}}(1+n_iD_i)(n_i+1)^A e^{-n_if(n_i)} + N\prod_{i\in\mathcal{G}}(n_i+1)^{|A|}e^{-n_if(n_i)}\right)$$

$$\leq (N+1) \sum_{\mathcal{G}\in 2^{[N]}\setminus\emptyset} \prod_{i\in\mathcal{G}} \sum_{n_i\in\mathbb{N}} (1+n_iD_i)(n_i+1)^A e^{-n_if(n_i)}$$

$$= \mathrm{O}(1). \quad \left(\text{by } n_if(n_i) = \Theta(n_i^{1/2})\right)$$

$\square$

*Proof of Lemma 10.* Because $\mathcal{A}(t)$, $(\mathcal{A}'(t))^c$ and $\mathcal{E}(t)$ imply

$$\|p^* - \hat{p}(t)\|_M \geq \sup_{p\in\mathcal{C}_1^c}\{\|p^* - p\|_M - \|\hat{p}(t) - p\|_M\}$$
$$\geq \sup_{p\in\mathcal{C}_1^c}\{\|p^* - \mathcal{C}_1^c\|_M - \|\hat{p}(t) - p\|_M\}$$
$$\geq \|p^* - \mathcal{C}_1^c\|_M - \nu_{1,L}\alpha(t)$$
$$\geq \|p^* - \mathcal{C}_1^c\|_M/2,$$

$(\mathcal{A}'(t))^c$, $\mathcal{D}(t)$ and $\mathcal{E}(t)$ imply

$$\max_i D(\hat{p}_i(t)\|S_ip^*) \geq 2\max_i \|\hat{p}_i(t) - S_ip^*\|^2$$
$$\geq 2\max_i (\|S_ip^* - S_i\hat{p}(t)\| - \|\hat{p}_i(t) - S_i\hat{p}(t)\|)_+^2$$
$$\geq 2\max_i \left(\|S_ip^* - S_i\hat{p}(t)\| - \sqrt{f(N_i(t))/4}\right)_+^2$$
$$\geq 2(\|p^* - \mathcal{C}_1^c\|_M/2 - \sqrt{\delta/2})_+^2$$
$$\geq \|p^* - \mathcal{C}_1^c\|_M^2/8. \quad \text{(by (23))}$$

On the other hand, event $\{J_i(t), \mathcal{A}^c(t), \mathcal{A}'(t), \min_j N_j(t) = n\}$ occurs for at most twice since all actions are put into the list if $\{\mathcal{A}^c(t), \mathcal{A}'(t)\}$ occurred. Thus, we have

$$\mathbb{E}\left[\sum_n \mathbb{1}\left[J_i(t), \mathcal{A}(t), (\mathcal{A}'(t))^c, \mathcal{D}(t), \mathcal{E}(t)\right]\right]$$

$$\leq 2\mathbb{E}\left[\sum_n \mathbb{1}\left[\bigcup_t \{\mathcal{A}(t), (\mathcal{A}'(t))^c, \mathcal{D}(t), \mathcal{E}(t), \min_j N_j(t) = n\}\right]\right]$$

$$\leq 2\sum_n \Pr\left[\max_j D(\hat{p}_j(t)\|S_jp^*) \geq \|p^* - \mathcal{C}_1^c\|_M^2/8, \bigcap_j\{N_j(t) \geq n\}\right]$$

$$\leq 2N\sum_n (n+1)^A e^{-n\|p^* - \mathcal{C}_1^c\|_M^2/8} \quad \text{(by (22))}$$

$$= \mathrm{O}(1).$$

$\square$

*Proof of Lemma 11.* Recall that

$$\mathcal{C}^c(t) = \left\{\{D(\hat{p}_{\hat{i}(t)}(t)\|S_{\hat{i}(t)}p^*) > f(N_{\hat{i}(t)}(t))\} \cup \bigcap_j\{D(\hat{p}_j(t)\|S_jp^*) \leq f(N_j(t))\}\right\}.$$

Here

$$\left\{ \mathcal{A}^c(t),\, \mathcal{D}(t),\, \mathcal{E}(t),\, \bigcap_j \{D(\hat{p}_j(t)\|S_j p^*) \le f(N_j(t))\} \right\} \tag{32}$$

cannot occur since (32) implies that

$$
\begin{aligned}
\|\hat{p}(t) - p^*\|_M &= \max_j \|S_j \hat{p}(t) - S_j p^*\| \\
&\le \max_j (\|S_j \hat{p}(t) - \hat{p}_j(t)\| + \|\hat{p}_j(t) - S_j p^*\|) \\
&\le \max_j \left( \sqrt{D(\hat{p}_j(t)\|S_j \hat{p}(t))/2} + \sqrt{D(\hat{p}_j(t)\|S_j p^*)/2} \right) \\
&\le \sqrt{2 \max_j f(N_j(t))} \quad \text{(by } \mathcal{D}(t)) \\
&\le 2\sqrt{\delta} \quad \text{(by (24))} \\
&\le \|p^* - \mathcal{C}_1^c\|_M / \sqrt{2} \quad \text{(by (23))} ,
\end{aligned}
$$

which contradicts $\hat{p}(t) \in \mathcal{C}_1^c$.

On the other hand, $\mathbb{1}\left[ J_i(n),\, \hat{i}(t) = j,\, D(\hat{p}_j(t)\|S_j p^*) > f(N_j(t)),\, N_j(t) = n_j \right]$ occurs for at most twice since $\hat{i}(t)$ is put into the list under this event. Thus, we have

$$
\begin{aligned}
\mathbb{E}\left[ \sum_{t=1}^{\infty} \mathbb{1}\left[ J_i(t),\, \mathcal{A}^c(t),\, \mathcal{C}^c(t),\, \mathcal{D}(t),\, \mathcal{E}(t) \right] \right] &\le 2 \sum_j \sum_{n=1}^{\infty} \Pr[D(\hat{p}_{j,\,n}\|S_j p^*) > f(n)] \\
&\le 2 \sum_j \sum_{n=1}^{\infty} (n+1)^A e^{-nf(n)} \quad \text{(by (22))} \\
&= \mathrm{O}(1) .
\end{aligned}
$$

$\square$

*Proof of Lemma 12.* $\mathcal{D}^c(t)$ implies

$$
\begin{aligned}
0 &< \min_p \sum_j N_j(t)(D(\hat{p}_j(t)\|S_j p) - f(N_j(t)))_+ \\
&\le \sum_j N_j(t)(D(\hat{p}_j(t)\|S_j p^*) - f(N_j(t)))_+
\end{aligned}
$$

and therefore

$$\bigcup_j \{D(\hat{p}_j(t)\|S_j p^*) \ge f(N_j(t))\} .$$

Note that $\{J_i(t),\, \mathcal{D}^c(t),\, N_j(t) = n\}$ occurs for at most twice because all actions are put into the list if $\mathcal{D}^c(t)$ occurred. Thus, we have

$$
\begin{aligned}
\mathbb{E}&\left[ \sum_n \mathbb{1}\left[ J_i(t),\, \mathcal{D}^c(t) \right] \right] \\
&\le 2\mathbb{E}\left[ \sum_j \sum_n \mathbb{1}\left[ D(\hat{p}_{j,n}\|S_j p^*) \ge f(n) \right] \right] \\
&\le 2 \sum_j \sum_n (n+1)^A e^{-nf(n)} \quad \text{(by (22))} \\
&= \mathrm{O}(1) .
\end{aligned}
$$

$\square$

*Proof of Lemma 13.* First, we have

$$\mathcal{E}^c(t) = \left\{ \max_i f(N_i(t)) > \min\left\{2\delta, (\rho_{1,L}\alpha(t))^2/4\right\} \right.$$

$$\cup \min_i N_i(t) < \max\{c\sqrt{\log t}, (\log\log T)^{1/3}\} \cup 2\nu_{1,L}\alpha(t) > \|p^* - \mathcal{C}_1^c\|_M \Big\}$$

$$\subset \left\{ f(\min_i N_i(t)) > \min\left\{2\delta, (\rho_{1,L}\alpha(t))^2/4\right\} \right.$$

$$\cup \min_i N_i(t) < c\sqrt{\log t} \cup c\sqrt{\log t} < (\log\log T)^{1/3} \cup 2\nu_{1,L}\alpha(t) > \|p^* - \mathcal{C}_1^c\|_M \Big\}$$

$$\subset \left\{ \frac{b}{\sqrt{c\sqrt{\log t}}} > \min\left\{2\delta, (\rho_{1,L}\alpha(t))^2/4\right\} \cup \min_i N_i(t) < c\sqrt{\log t} \right.$$

$$\cup\ t < e^{\frac{(\log\log T)^{2/3}}{c}} \cup 2a/\log\log t > \|p^* - \mathcal{C}_1^c\|_M/\rho_{1,L} \Big\}$$

$$= \left\{ t < e^{\frac{b^4}{16\delta^4 c^2}} \cup \frac{(\log t)^{1/4}}{\log\log t} < \frac{b}{a\sqrt{c}\rho_{1,L}} \cup \min_i N_i(t) < c\sqrt{\log t} \right.$$

$$\cup\ t < e^{\frac{(\log\log T)^{2/3}}{c}} \cup t < e^{e^{2a\rho_{1,L}/\|p^* - \mathcal{C}_1^c\|_M}} \Big\}.$$

From $\lim_{t\to\infty}(\log t)^{1/4}/\log\log t = \infty$ and $e^{\frac{(\log\log T)^{2/3}}{c}} = o(e^{\log\log T}) = o(\log T)$ we have

$$\sum_{t=1}^{T} \mathbb{1}\left[J_i(t), \mathcal{E}^c(t)\right]$$

$$= \sum_j \sum_{t=1}^{T} \mathbb{1}\left[J_i(t), N_j(t) < c\sqrt{\log t}\right] + o(\log T).$$

By (8), event $\{J_i(t), N_j(t) < c\sqrt{\log t}, N_j(t) = n\}$ occurs for at most twice and therefore

$$\sum_{t=1}^{T} \mathbb{1}\left[J_i(t), \mathcal{E}^c(t)\right]$$

$$\leq 2\sum_j \sum_{n=1}^{T} \mathbb{1}\left[\bigcup_{t=1}^{T}\{n < c\sqrt{\log t}\}\right] + o(\log T)$$

$$= o(\log T).$$

$\square$