[Reviews · NeurIPS 2015]

Submitted by Assigned_Reviewer_1

This paper considers finite partial monitoring problems, and considers distribution dependent regret bounds for them. It first proves a logarithmic lower bound on the regret of any "strongly consistent" algorithm with a precise characterization of the constant in front of the logarithmic term. The paper then introduces two algorithms: PM-DMED and PM-DMED-Hinge, the former of which empirically matches the regret lower bound, but lacks a theoretical analysis, and the latter which has a provable regret bound matching the lower bound (up to the constant term) asymptotically.

While the lower bound is based on pretty standard arguments, the upper bound analysis is somewhat novel. The paper is quite well-written and has nice results, and extends the work done in the multiarmed bandits problem on deriving distribution dependent logarithmic regret bounds to the partial monitoring setting.
Summary: The main contribution of this paper is deriving an algorithm for stochastic partial monitoring problems with a logarithmic regret bound that matches a lower bound proved in the same paper up to the constant in front of log(T). While the logarithmic lower bound is quite standard, the algorithm matching the lower bound is quite novel.

Submitted by Assigned_Reviewer_2

This paper studies the stochastic partial monitoring problem. Asymptotic lower bound is derived and an asymptotically optimal algorithm is proposed.

Overall Comments:

- The lower bound might be easier to be derived from the results in the following paper: Graves, Todd L., and Tze Leung Lai. "Asymptotically efficient adaptive choice of control laws incontrolled markov chains." SIAM journal on control and optimization 35.3 (1997): 715-743.

- I did not check the proof of the optimality for the algorithm but I think the result is reasonable: The lower bound requires a solution to an optimization problem which represents how many times each action needs to be played to eliminate "confusing environments". The algorithm tries to approximate these number of plays by the empirical estimate of p. A uniform exploration rate (\sqrt(log t)) is used the guarantee the approximation quality of the solution to the optimization problem.

Quality, clarity, originality and significance:

The paper is well written and the contribution of asymptotic analysis is interesting.

The existing CBP algorithm for stochastic partial monitoring is designed for achieving the T^2/3 minimax bound and does not exploit the observation structure efficiently and thus is not asymptotically optimal. It could be more interesting to consider designing an algorithm which is both asymptotically and minimax optimal but this might be harder and beyond the scope of this paper.

I am not very sure about the significance. Although the ideas of deriving the lower bound and the algorithm are not completely novel fitting these ideas into the partial monitoring framework still require some efforts.
Summary: This paper provides interesting contribution to the stochastic partial monitoring problem.

Submitted by Assigned_Reviewer_3

The paper presents distribution dependent bounds for the stochastic partial monitoring problem. The main result is a lower bound that is valid for all globally observable problem instances (i.e., instances with sublinear worst case regret bound). The tightness of this result is also shown: the paper presents an algorithmic solution and an upper bound on its regret that matches the lower bound. Finally, the authors demonstrate through experiments the superiority of these algorithm over previous ones.

The paper is very interesting, and the results nicely complement the existing minimax bounds. The bounds are based on a novel complexity notion, which is somewhat complicated, but is reasonable. The writing style is very clear, and the details are elaborated in a thorough way. All in all, a very nice paper.

Some questions and comments: - How does the lower bound compare to the one in [11] (in case of an easy instance)? - The notation ||.||_M is rather confusing. It doesn't seem to be any different from the squared norm. - Denoting the learners strategy by q and the opponents one by p concisely would be helpful for the reader. - The goal of the last paragraph in Section 4 is to explain the \sqrt{log T} exploration. However, it is still not clear why the exploration corresponding to Equations (7) and (8) does not suffice. Additionally, can, maybe, a concentration result for the KL-divergence (used in (5)) replace all of (6), (7) and (8)?

Typos: l130: C_1^c should be C_i^c l132: S_i has dimension KxM (and not AxM) l615: Is q'_{t_u} correct? Shouldn't it be q'_{T_{t_u}}?
Summary: The paper presents distribution dependent bounds for the stochastic partial monitoring problem. The paper is very interesting, and the results nicely complement the existing minimax bounds.

Submitted by Assigned_Reviewer_4

What do the quantities defined in Equations (2), (3) and (4) intuitively mean?

Why the regret of some of the approaches in ``Figure (c) three-states harsh" is less than the theoretical lower bound? Is it because t is not large enough?
Summary: The authors propose a distribution dependent lower bound on the regret of some class of stochastic partial monitoring problem. Then they propose an algorithm that seems to achieve this lower bound in practice although the proofs are lacking.

Author Feedback
Author rebuttal: We thank you for your time and many insightful comments.

Reviewer 1:
Let us clarify the novelty of the regret upper bound. The bound in Thm 3 is new in the following two senses: (1) a logarithmic regret bound for hard partial monitoring problems. Note that, existing algorithms such as CBP [11] has a logarithmic regret bound only for easy problems. (2) The logarithmic regret bound for easy and hard problems with its constant factor in front of log T asymptotically matches the lower bound.

Reviewer 2:
>The lower bound might be easier to be derived from the results in [Graves&Lai 97]
Thank you for a reference. The paper is an extension of [Lai&Robbins 1985] for some class of sequential optimizations such as K-armed bandits and unimodal bandits. Although the expression of the regret lower bound by [Graves&Lai] is similar to the one of our lower bound in partial monitoring (PM), we do not think that PM fits into their framework: in Graves&Lai s' framework, reward r(Xt, ut) must be a function of observation Xt and action ut. Unlike bandits, some signals in PM are not informative enough to determine the amount of reward, and thus we cannot define the reward as a function of the observation and the action.

Reviewer 3:
>How does the lower bound compare to the one in [11] (in case of an easy instance)?
CBP has a logarithmic a regret upper bound (Thm 1 in [11]). The regret analysis there is based on the locally observable structure: in particular, term d_k in [11] on the constant factor is based on the path argument as Fig 1 [11], which is O(N) in many cases. In that cases, the leading logarithmic term of Thm 1 in [11] can be cubic in N. Whereas our optimal bound is at most linear dependence on N.
>The notation ||.||_M is rather confusing. It doesn't seem to be any different from the squared norm.
Sorry for the confusion. In our paper, ||.|| is the standard 2-norm, whereas ||.||_M is the maximum over the 2-norms in view of the observation from each action i, which is defined in the beginning of Appendix D. We will make this clear at the first appearance of the notation in the main paper.
> It is still not clear why the exploration corresponding to Equations (7) and (8) does not suffice.
Terms (6)(7)(8) are necessary for the following reason: Term (5) is a kind of maximum likelihood estimation (MLE) that integrates the observations from all actions. Terms (7)(8) are necessary for calculating the minimum amount of exploration that is defined in (2)-(4) in Sec 3, and is based on the assumption that MLE is close to the true p^*. Thanks to the \sqrt{log T} exploration in (6), this assumption holds with a high probability.
>l132: S_i has dimension KxM (and not AxM)
K is not defined in our paper (you might mean N). S_i is a linear operator that maps p (distribution over outcomes) into S_i p (distribution over symbols under action i). Therefore, S_i is R^{AxM}. Previous papers such as [11] defined S_i of dimension (s_i)xM, where s_i is the number of symbols observable under action i. This version of S_i maps p to the space of observable symbols by selecting action i, which essentially conveys the same information as our ones.
>l615: Is q'_{t_u} correct? Shouldn't it be q'_{T_{t_u}}?
It is correct since we defined q'_t only for rounds T_t, t=1,2,.., in l610. If we define q'_{T_t} instead of q'_t in l610 then q'_{t_u} in l615 becomes q'_{T_{t_u}} but we think that current notation is concise.

Reviewer 5:
We admit that the assumptions are a little bit involved, but we think that they come from the inherent difficulty of our goal to optimize the constant factor of the log(T) term. In fact, if we just want a logarithmic regret bound then we can remove the assumptions by using Lem 5, which gives an upper bound of R_j^* without any assumption.

Reviewer 6:
>proofs are lacking
We derived an asymptotically optimal regret upper bound of PM-DMED-Hinge for trivial, easy and hard partial monitoring problems (Thm 3). PM-DMED, a simplified version, has no regret bound, but its regret empirically matches the lower bound for all simulations in the paper.
>What do the quantities defined in Equations (2), (3) and (4) intuitively mean?
These quantities characterize the possible minimum regret: it is necessary to explore each arm i at least r_i log T times for some {r_i} in R_1 (term (2)) to satisfy the strong consistency. C_1^* logT in (3) corresponds to the minimum regret such that (2) is satisfied. The set of solutions (number of draws) that achieves the minimum regret of (3) is r_i^* log T for some {r_i^*} \in R_1^* (term (4)). We will make this clear.
>Why the regret of some of the approaches in some Figs is less than the theoretical lower bound?
This is because, only an asymptotic bound on the logarithmic term is derived and it does not mention the constant term. In bandit problems, the regret of algorithms is often smaller than this line, which is reported in some papers, such as Figs 1 and 2 in [17].